# Discovering Data Structures: Nearest Neighbor Search and Beyond

## Abstract

We propose a general framework for end-to-end learning of data structures. Our framework adapts to the underlying data distribution and provides fine-grained control over query and space complexity. Crucially, the data structure is learned from scratch, and does not require careful initialization or seeding with candidate data structures/algorithms. We first apply this framework to the problem of nearest neighbor search. In several settings, we are able to reverse-engineer the learned data structures and query algorithms. For 1D nearest neighbor search, the model discovers optimal distribution (in)dependent algorithms such as binary search and variants of interpolation search. In higher dimensions, the model learns solutions that resemble k-d trees in some regimes, while in others, elements of locality-sensitive hashing emerge. The model can also learn useful representations of high-dimensional data and exploit them to design effective data structures. We also adapt our framework to the problem of estimating frequencies over a data stream, and believe it could also be a powerful discovery tool for new problems.

## 1 Introduction

*Can deep learning models be trained to discover data structures from scratch?*

There are several motivations for this question. The first is scientific. Deep learning models are increasingly performing tasks once considered exclusive to humans, from image recognition and mastering the game of Go to engaging in natural language conversations. Designing data structures and algorithms, along with solving complex math problems, are particularly challenging tasks. They require searching through a vast combinatorial space with a difficult to define structure. So, it is natural to ask what it would take for deep learning models to solve such problems. There are already promising signs: these models have discovered fast matrix-multiplication algorithms (Fawzi et al., 2022), solved SAT problems (Selsam et al., 2018), and learned optimization algorithms for learning tasks (Garg et al., 2022; Akyürek et al., 2022; Fu et al., 2023; Von Oswald et al., 2023). In this work, we investigate the problem of data structure discovery, with a focus on nearest neighbor search.

The second motivation is practical. Data structures are ubiquitous objects that enable efficient querying. Traditionally, they have been designed to be worst-case optimal and therefore agnostic to the underlying data and query distributions. However, in many applications there are patterns in these distributions that can be exploited to design more efficient data structures. This has motivated recent work on learning-augmented data structures which leverages knowledge of the data distribution to modify existing data structures with predictions (Lykouris & Vassilvitskii, 2018; Ding et al., 2020; Lin et al., 2022a; Mitzenmacher & Vassilvitskii, 2022). In much of this work, the goal of the learning algorithm is to learn distributional properties of the data, while the underlying query algorithm/data structure is hand-designed. Though this line of work clearly demonstrates the potential of leveraging distributional information, it still relies on expert knowledge to incorporate learning into such structures. In our work, we ask if it is possible to go one step further and let deep learning models discover entire data structures and query algorithms in an end-to-end manner.

### 1.1 Framework for data structure discovery

Data structure problems can often be decomposed into two steps: 1) data structure construction and 2) query execution. The first step transforms a raw dataset $D$ into a structured database $\hat{D}$, while

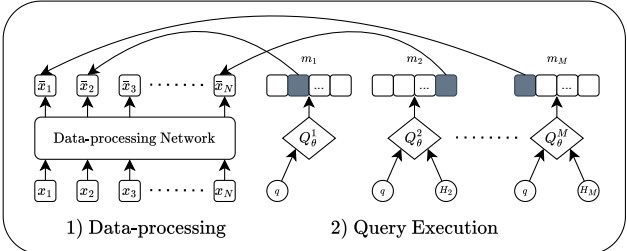

Figure 1: Our model has two components: **1)** A data-processing network that transforms raw data into structured data, arranging it for efficient querying and generating additional statistics when given extra space (not shown). **2)** A query-execution network that performs $M$ lookups into the output of the data-processing network in order to retrieve the answer to some query $q$. Each lookup $i$ is managed by a separate query model $Q_\theta^i$, which takes $q$ and the lookup history $H_i$, and outputs a one-hot lookup vector $m_i$ indicating the position to query.

query-execution performs lookups into $\hat{D}$ to retrieve the answer for some query $q$. The performance of a data structure is typically quantified in terms of two measures: *space complexity*—how much memory is required to store the data structure, and *query complexity*—how many lookups into the data structure are required to answer some query. One can typically tradeoff larger space complexity for smaller query complexity, and vice versa. We focus on these criteria as they are widely studied and have clear practical connections to efficiency.

To learn such data structures, we have a *data-processing network* which learns how to map the dataset to the data structure, and a *query network* which learns an algorithm for using the data structure to answer queries (Figure 1). In order to learn efficient data structures and query algorithms we impose constraints on the size of the data structures and on the number of lookups that the query network can make into the data structure. Crucially, we propose end-to-end training of both networks such that the learned data structure and query algorithm are optimized for one another. Moreover, in settings where it is beneficial to learn lower-dimensional representations from high-dimensional data, E2E training encourages the representations to better capture features of the problem that the data structure can exploit.

On the one hand, learning this data-processing network and query network jointly, end-to-end, seems obvious — especially given the many successes of end-to-end learning over the past decade. On the other hand, it might be hard to imagine such learning getting off the ground. For instance, if the data-processing network produces a random garbled function of the dataset, the query model cannot hope to do anything meaningful. This is further complicated by the fact that these data structure tasks are more discrete and combinatorial in terms of how the query model accesses the data structure.

### 1.2 SUMMARY OF RESULTS

We apply this framework to the problem of nearest neighbor (NN) search in both low and high dimensions. Given the extensive theoretical work on this topic, along with its widespread practical applications, NN search is an ideal starting point for understanding the landscape of end-to-end data structure discovery. Beyond NN search, we explore the problem of frequency estimation in streaming data and discuss other potential applications of this framework. Our findings are:

**Sorting and searching in 1D**   For 1D NN search, the data-processing network learns to sort, while the query network simultaneously learns to search over the sorted data. When the data follows a uniform or Zipfian distribution, the query network exploits this structure to outperform binary search. On harder distributions lacking structure, the network adapts by discovering binary search, which is worst-case optimal. Importantly, the model discovers that sorting followed by the appropriate search algorithm is effective for NN search in 1D without explicit supervision for these primitives.

**K-d trees in 2D**   In 2D, when the data is drawn from a uniform distribution, the model discovers a data structure that outperforms k-d trees. On harder distributions, the learned structure shows surprising resemblance to a k-d tree. This is striking as a k-d tree is a non-trivial data structure, constructed by recursively partitioning the data and finding the median along alternating dimensions.

**Useful representations in high dimensions** For high-dimensional data, the model learns representations that make NN search efficient. For example, with data from a uniform distribution on a 30-dimensional hypersphere, the model partitions the space by projecting onto a pair of vectors, similar to locality-sensitive hashing. When trained on an extended 3-digit MNIST dataset, the model finds 1D features that capture the relative ordering of the digits, sorts the images using these features, and performs a search on the sorted images—all of which is learned jointly from scratch.

**Trading off space and query efficiency** An ideal data structure should be able to use extra space to improve query efficiency by storing additional statistics. The learned model demonstrates this behavior, with performance improving monotonically as more space is provided, in both low and high dimensions. Thus, the model learns to effectively trade off space for query efficiency.

**Beyond NN search** We also explore the classical problem of frequency estimation, where a memory-constrained model observes a stream of items and must approximate the frequency of a query item. The learned structure exploits the underlying data distribution to outperform baselines like CountMin sketch, demonstrating the broader applicability of the framework beyond NN search.

## 2 NEAREST NEIGHBOR SEARCH

Given a dataset $D = \{x_1, ..., x_N\}$ of $N$ points where $x_i \in \mathbb{R}^d$ and a query $q \in \mathbb{R}^d$, the nearest neighbor $y$ of $q$ is defined as $y = \arg\min_{x_i \in D} \ dist(x_i, q)$. We focus on the case where $dist(\cdot)$ corresponds to the Euclidean distance. Our objective is to learn a data structure $\hat{D}$ for $D$ such that given $q$ and a budget of $M$ lookups, we can output a (approximate) nearest neighbor of $q$ by querying at most $M$ elements in $\hat{D}$. When $M \geq N$, $y$ can be trivially recovered via linear search so $\hat{D} = D$ is sufficient. Instead, we are interested in the case when $M \ll N$.[1]

### 2.1 SETUP

**Data-processing Network** Recall that the role of the data-processing network is to transform a raw dataset into a data structure. The backbone of our data-processing network is an 8-layer transformer model based on the NanoGPT architecture (Karpathy, 2024). In the case of NN search, we want the data structure to preserve the original inputs and just reorder them appropriately as the answer to the nearest neighbor query should be one of elements in the dataset. Therefore, the transformer takes as input the dataset $D$ and outputs a scalar $o_i \in \mathbb{R}$ representing the rank for each point $x_i \in D$. These rankings $\{o_1, ..., o_N\}$ are then sorted using a differentiable sort function, $sort(\{o_1, o_2 \ldots, o_N\})$ (Grover et al., 2019; Cuturi et al., 2019; Petersen et al., 2022), which produces a permutation matrix $P$ that encodes the order based on the rankings. By applying $P$ to the input dataset $D$, we obtain $\hat{D}_P$, where the input data points are arranged in order of their rankings. By learning to rank rather than directly outputting the transformed dataset, the transformer avoids the need to reproduce the exact inputs. Note that this division into a ranking model followed by sorting is without loss of generality as the overall model can represent any arbitrary ordering of the inputs.

We also consider scenarios where the data structure can use additional space. To support this use case, the transformer can also output $T$ extra tokens $b_1, ..., b_T \in \mathbb{R}^d$ which can be retrieved by the query-execution network. We form the data structure $\hat{D}$ by concatenating the permuted inputs and the extra tokens: $\hat{D} = [\hat{D}_P, b_1, ..., b_T]$.

**Query Execution Network** The role of the query-execution network is to output a (approximate) nearest-neighbor of some query $q$ given a budget of $M$ lookups into the data structure $\hat{D}$. The query-execution network consists of $M$ MLP query models[2] $Q^1_{\theta_1}, ..., Q^M_{\theta_M}$. Each query model $Q^i_{\theta_i}$ outputs a one-hot vector $m_i \in \mathbb{R}^{N+T}$ which represents a lookup position in $\hat{D}$. To execute the lookup, we compute the value $v_i$ at the position denoted by $m_i$ in $\hat{D}$ as $v_i = m_i^\top \hat{D}$. In addition to the query $q$, each query model $Q^i_{\theta_i}$ also takes as input the query execution history $H_i = \{(m_1, v_1), ..., (m_{i-1}, v_{i-1})\}$ where $H_1 = \emptyset$. The final answer of the network for the nearest-neighbor query is given by $\hat{y} = m_M^\top \hat{D}$.

To restrict our model to exactly $M$ lookups, we enforce each lookup vector $m_i$ to be a one-hot vector. Enforcing this constraint during training poses a challenge as it is a non-differentiable operation.

---

[1] E.g. in 1D, binary search requires $M = \log(N)$ lookups given a sorted list.

[2] The query models do not share weights.

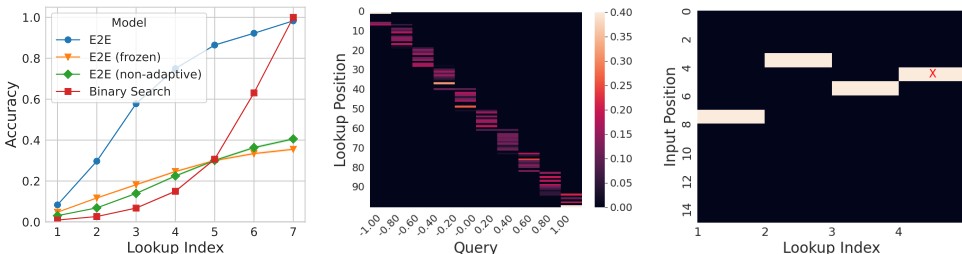

Figure 2: **(Left)** Our model (E2E) trained with 1D data from the uniform distribution over $(-1, 1)$ outperforms binary search and several ablations. **(Center)** Distribution of lookups by the first query model. Unlike binary search, the model does not always start in the middle but rather closer to the query's likely position in the sorted data. **(Right)** When trained on data from a "hard" distribution for which the query value does reveal information about the query's relative position, the model finds a solution similar to binary search. The figure shows an example of the model performing binary search ('X' denotes the nearest neighbor location).

Instead, during training, our model outputs soft-lookups where $m_i$ is the output of the softmax function and $\sum_j m_{ij} = 1$. This alone, however, leads to non-sparse queries. To address this, we add noise to the logits before the softmax operation (only during training). This regularizes the query network, encouraging it to produce sparser solutions (see App C.1 for details as to why this occurs). Intuitively, the network learns a function that is robust to noise, and the softmax output becomes robust when the logits are well-separated. Well-separated logits, in turn, lead to sparser solutions.

**Data Generation and Training** Each training example is a tuple $(D, q, y)$ consisting of a dataset $D$, query $q$, and nearest neighbor $y$ generated as follows: (i) sample dataset $D = \{x_1, ..., x_N\}$ from dataset distribution $P_D$, (ii) sample query $q$ from query distribution $P_q$, (iii) compute nearest neighbor $y = \arg\min_{x_i \in D} ||x_i - q||_2$. The dataset and query distributions $P_D, P_q$ vary across the different settings we consider and are defined later. Given a training example $(D, q, y)$, the data-processing network transforms $D$ into the data structure $\hat{D}$. Subsequently, the query-execution network, conditioned on $q$, queries the data structure to output $\hat{y}$. We use SGD to minimize either the square loss $||\hat{y} - y||_2^2$ or the cross-entropy loss between $\hat{y}$ and $y$ averaged over all training examples. This is an empirical choice, and in some settings one loss function performs better than the other. After training, we test our model on inputs $(D, q, y)$ generated in the same way. We describe the exact model architecture and training hyper-parameters in App A.1.

## 2.2 EXPERIMENTS

We evaluate our end-to-end model (referred to as *E2E*) on one-dimensional, two-dimensional, and high-dimensional nearest-neighbor problems. We primarily focus on data structures that do not use extra space, but in Section 2.2, we also explore scenarios with additional space.

**Baselines** We compare against suitable NN data structures in each setting (e.g., sorting followed by binary search in 1D), and also against several ablations to study the impact of various model components. The *E2E (frozen)* model does not train the data-processing network, relying on rankings generated by the initial weights. The *E2E (no-permute)* model removes the permutation component of the data-processing network so that the transformer has to learn to transform the data points directly. The *E2E (non-adaptive)* model ablation conditions each query model $Q_{\theta_i}^i$ on only the query $q$ and not the query history $H_i$.

### ONE-DIMENSIONAL DATA

**Uniform Distribution** We consider a setting where the data distribution $P_D$ and query distribution $P_q$ correspond to the uniform distribution over $(-1, 1)$, $N = 100$ and $M = 7$. We plot the accuracy[3], which refers to zero-one loss in identifying the nearest neighbor, after each lookup in Figure 2 (Left). Recall that $v_i$ corresponds to the output of the $i$-th lookup. Let $v_i^*$ denote the closest element to the

---

[3]We include MSE plots as well in App. B.

query so far: $v_i^* = \arg\min_{v \in \{v_1,...,v_i\}} ||v - q||_2^2$. At each lookup index we plot the nearest neighbor accuracy corresponding to $v_i^*$. We do this for all the methods.

A key component in being able to do NN search in 1D is sorting. We observe that the trained model does indeed learn to sort. We verify this by measuring the fraction of inputs that are mapped to the correct position in the sorted order, averaged over multiple datasets. The trained model correctly positions approximately 99.5% of the inputs. This is interesting as the model never received explicit feedback to sort the inputs and figured it out in the end-to-end training. The separate sorting function aids the process, but the model still had to learn to output the correct rankings.

The second key component is the ability to search over the sorted inputs. Here, our model learns a search algorithm that outperforms binary search, which is designed for the worst case. This is because unlike binary search, our model exploits knowledge of the data distribution to start its search closer to the nearest neighbor, similar to interpolation search (Peterson, 1957). For instance, if the query $q \approx 1$, the model begins its search near the end of the list (Figure 2 (Center)). The minor sorting error ($\sim 0.5\%$) our model makes likely explains its worse performance on the final query.

To understand the relevance of different model components, we compare against various ablations. The *E2E (frozen)* model (untrained transformer) positions only about 9% of inputs correctly, explaining its under-performance. This shows that the transformer must learn to rank the inputs appropriately, and that merely using a separate function for sorting the transformer output is insufficient. The *E2E (non-adaptive)* baseline, lacking query history access, underperforms as it fails to learn adaptive solutions crucial for 1D NN search. The *E2E (no-permute)* ablation (Figure 9 (Left))[4] struggles to fully retain inputs. We verify this by measuring the distance between the transformer's inputs and outputs. These ablations highlight the crucial role of both learned orderings and query adaptivity for our model.

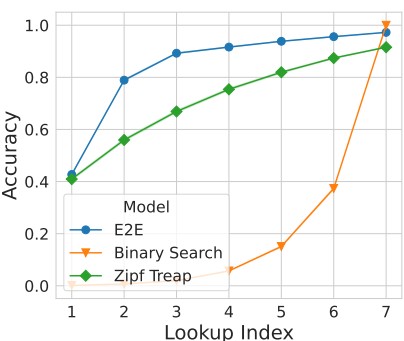

Figure 3: For a 1D Zipfian query distribution, our model significantly outperforms binary search and is competitive with the learning-augmented treap algorithm from Hsu et al. (2019)

**Zipfian Distribution**  Prior work has shown that several real-world query distributions follow a Zipfian trend whereby a few elements are queried far more frequently than others, leading to the development of learning-augmented algorithms aimed at exploiting this (Lin et al., 2022b). We consider a setting where $P_D$ is the discrete uniform distribution over $\{1, ..., 200\}$ and $P_q$ is a Zipfian distribution over $\{1, ..., 200\}$ skewed towards smaller numbers such that the number $i$ is sampled with probability proportional to $\frac{1}{i^\alpha}$. We set $\alpha = 1.2$. Again, in this setting $N = 100$ and $M = 7$.

In Figure 3 we compare our model to both binary search and the learning-augmented treap from Lin et al. (2022a). Our model performs slightly better than the learning-augmented treap and both algorithms significantly outperform binary search with less than $\log(N)$ queries as expected. This result highlights a crucial difference in spirit between our work and much of the existing work on learning-augmented algorithms. While the Zipfian treap incorporates learning in the algorithm, the authors still had to figure out how an existing data structure could be modified to support learning. On the other hand, by learning end-to-end, our framework altogether removes the need for the human-in-the-loop. This is promising as it could be useful in settings where we lack insight on appropriate data structures. The flip side, however, is that learning-augmented data structures usually come with provable guarantees which are difficult to get when training models in an end-to-end fashion.

**Hard Distribution**  To verify that our model can also learn worst-case optimal algorithms such as binary search, we design a hard distribution $\mathcal{D}_\mathcal{H}$ with the property that for any given query there does not exist a strong prior over the position of its nearest neighbor in the sorted data (see App. B.1 for more details about $\mathcal{D}_\mathcal{H}$). We generate our queries by first sampling a point (uniformly at random)

---

[4]We only measure MSE for this baseline. Given that it does not preserve inputs we cannot measure accuracy.

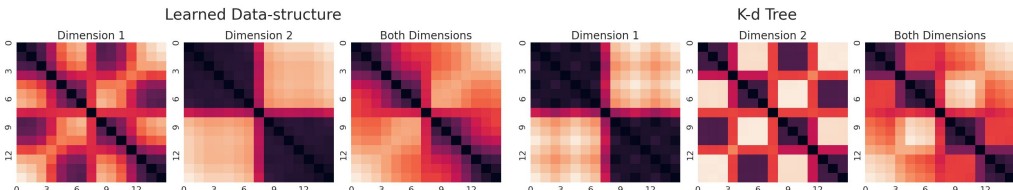

Figure 4: The learned data structure resembles a k-d tree in 2D. We show the average pairwise distances (across the first, second and both dimensions) between points at different positions for the learned data structure and k-d tree, with darker colors indicating smaller distances. For the k-d tree, data is arranged by in-order traversal of the tree. The plots look similar for k-d trees and the learned data structure, with dimensions 1 and 2 flipped. The k-d tree first splits all points into two groups by the first axis which is why points 0-7 are on average closer to one another than the others along dimension 1. The k-d tree then partitions each group by the second axis. This plot demonstrates that our model does something similar, first splitting by the second axis and then the first.

from the dataset and then adding noise from the standard normal distribution. We use $N = 15$ and $M = 3$. Since the hard distribution generates numbers at several scales, this makes it challenging to train the model with larger $N$. In general, we find that training models is easier when there is more structure in the distribution to be exploited.

The model does indeed discover a search algorithm similar to binary search. In Figure 2 (Right), we show a representative example of the model's search behavior, resembling binary search (see Figure 14 for more examples). The error curve in Figure 12 also closely matches that of binary search.

In summary, in all the above settings, starting from scratch, the data-processing network discovers that the optimal way to arrange the data is in sorted order. Simultaneously, the query-execution network learns to efficiently query this sorted data, leveraging the properties of the data distribution.

TWO-DIMENSIONAL DATA

Beyond one dimension it is less clear how to optimally represent a collection of points as there is no canonical notion of sorting along multiple dimensions. In fact, we observe in these experiments that different data/query distributions lead to altogether different data structures. This reinforces the value in learning both the data structure and query algorithm together end-to-end.

**Uniform Distribution**   We use a similar setup to 1D, sampling both coordinates independently from the uniform distribution on $(-1, 1)$. We set $N = 100$ and $M = 6$, and compare to a k-d tree baseline. Our E2E model has a NN accuracy of $75\%$ vs $52\%$ for the k-d tree (Fig. 8 in App. B). A k-d tree is a binary tree for organizing points in k-dimensional space, with each node splitting the space along one of the k axes, cycling through the axes at each tree level. Our model outperforms the k-d tree as it can exploit distributional information. By studying the permutations, we find that our model learns to put points that are close together in the 2D plane next to each other in the permuted order (see Fig. 16 for an example).

**Hard Distribution**   We also consider the case where we sample both coordinates independently from the hard distribution considered in the 1D setup (see Figure 15 for the corresponding error curve). We observe that the data structure learned by our model is surprisingly similar to a k-d tree (see Fig 4). This is striking as a k-d tree is a non-trivial data structure, requiring recursively partitioning the data and finding the median along alternating dimensions at each level of the tree.

HIGH-DIMENSIONAL DATA

High-dimensional NN search poses a challenge for traditional low-dimensional algorithms due to the curse of dimensionality. K-d trees, for instance, can require an exponential number of queries in high dimensions (Kleinberg, 1997). This has led to the development of approximate NN search methods such as locality sensitive hashing (LSH) which have a milder dependence on $d$ (Andoni et al., 2018), relying on hash functions that map closer points in the space to the same hash bucket.

We train our model on datasets uniformly sampled from the $d$-dimensional unit hypersphere. The query is sampled to have a fixed inner-product $\rho \in [0, 1]$ with a dataset point. When $\rho = 1$, the

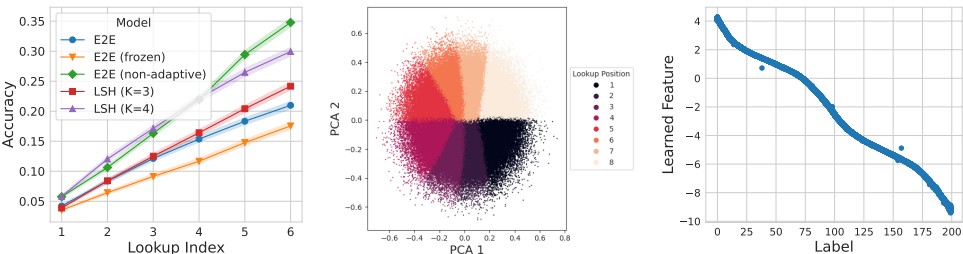

Figure 5: **(Left)** For NN search in higher dimensions (d = 30), the trained models perform comparably to (E2E) or better than (E2E (non-adaptive)) locality-sensitive hashing (LSH) baselines. **(Center)** When trained with a single query, the model partitions the query space based on projection onto two vectors, similar to LSH. We show the query projection onto the subspace spanned by these vectors and the lookup positions for different queries. **(Right)** When trained end-to-end to do nearest neighbor search over 3-Digit MNIST Images, our model learns 1D features that capture the relative ordering of the numbers in the images.

query matches a data point, making hashing-based methods sufficient. For $\rho < 1$, LSH-based solutions are competitive. We train our model for $\rho = 0.8$ and compare it to an LSH baseline when $N = 100, M = 6$, and $d = 30$. In Figure 5 (Left), we observe that our model performs competitively with LSH baselines (see details of the baselines in App D).[5] The non-adaptive model does slightly better as adaptivity is not needed to do well here (e.g., LSH is non-adaptive), and lack of adaptivity likely makes training easier. To better understand the data structure our model learns we consider a smaller setting where $N = 8$ and $M = 1$. We find that the model learns an LSH like solution, partitioning the space by projecting onto two vectors in $\mathbb{R}^{30}$ (see Figure 5 (Center)). We provide more details in App C.3.

**Learning useful representations** High-dimensional data often contains low-dimensional structures, such as data lying on a manifold, which can be leveraged to improve the efficiency of NN search. ML models are better suited than humans to exploit these structures. We investigate whether our end-to-end learning framework can learn such structures. This is a challenging task as it involves jointly optimizing the learned representation, data structure, and query algorithm.

We consider the following task: given a dataset of distinct 3-digit handwritten number images, and a query image, find its nearest neighbor in the dataset, which corresponds to the image encoding the same number as the query image (i.e., *nearest* is defined over the label space).

We generate images of 3-digit numbers by concatenating digits from MNIST (see Figure 11 for image samples). To construct a nearest-neighbor dataset $D$, we sample $N = 50$ labels (each label corresponds to a number) uniformly from 0 to 200. For each label, we then sample one of its associated training images from 3-digit MNIST. Additionally, we sample a query label (uniformly over $\{1, .., 200\}$) and its corresponding training image and find its nearest neighbor in $D$, which corresponds to the image with the same label. We emphasize that the model has no label supervision but rather only has access to the query's nearest neighbor. After training, we evaluate the model using the same data generation process but with images sampled from the 3-digit MNIST test set.

As both the data-processing and query-execution networks should operate over the same low-dimensional representation we train a CNN feature model $F_\phi$ as well. Our setup remains the same as before except now the data-processing network and query-execution network operate on $\{F_\phi(x_1), ..., F_\phi(x_N)\}$ and $F_\phi(q)$, respectively. As the model does not have access to the underlying metric space we minimize the cross-entropy loss instead of the MSE.

Ideally, the feature model $F$ should learn 1d features which encode the relative ordering of the numbers, the data model sorts them, and the query model should do some form of interpolation search where it can use the fact that the data distribution is uniform to do better than binary search. This is almost exactly what all models learn to do, from scratch, in an end-to-end fashion, without any explicit supervision about which image encodes which number. In Figure 5 (Right) we plot the

---

[5]We exclude LSH baselines with larger K as they under-perform.

learned features of the model. We find that the data model learns to sort the features with 98% accuracy and the query model finds the nearest neighbor with almost 100% accuracy (Figure 10).

LEVERAGING EXTRA SPACE

The previous experiments demonstrate our model's ability to learn useful orderings for efficient querying. However, data structures can also store additional pre-computed information to speed up querying. For instance, with infinite extra space, a data structure could store the nearest neighbor for every possible query, enabling $O(1)$ search. Here, we evaluate if our model can effectively use extra space.

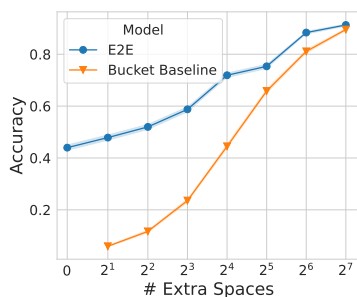

We run an experiment where the data and query distribution are uniform over $(-1, 1)$ with $N = 50, M = 2$. We allow the data-processing network to output $T \in \{0, 2^1, 2^2, 2^3, 2^4, 2^5, 2^6, 2^7\}$ tokens $b_1, ..., b_T \in \mathbb{R}$ in addition to the $N$ rankings. We plot the NN accuracy as a function of $T$ in Figure 5 (Right) compared to a simple bucketing baseline that partitions $[-1, 1]$ into $T$ evenly-sized buckets and in each bucket stores $\arg\min_{x_j \in D} ||x_j - l_i||$ where $l_i$ is the midpoint of the segment corresponding to bucket $i$ (described fully in App C.4.1). The accuracy monotonically increases with extra

Figure 6: For NN search in 1D the model learns to use extra space and outperforms a bucketing baseline.

space demonstrating that the data-processing network learns to pre-compute useful statistics that enable more efficient querying. We provide some insight into the learned solution in App C.4 and show how our model can be trained to use extra space in 30D as well (App C.5).

## 3 BEYOND NEAREST NEIGHBOR SEARCH

Many other data structure problems beyond nearest neighbor search can be modeled by our framework. Here, we illustrate this broader applicability by applying the framework to the classical problem of *frequency estimation*: a memory-constrained model observes a stream of elements, and is subsequently asked to approximate the number of times a query element has appeared (Cormode & Muthukrishnan, 2005; Cormode & Hadjieleftheriou, 2010). In Section 3.2 we describe several other data structure problems that our framework could be applied to.

### 3.1 FREQUENCY ESTIMATION

Given a sequence of $T$ elements $e_i^{(1)}, ..., e_j^{(T)}$, the task is to estimate the frequency of a query element $e_q$ up until time-step $T$. Specifically, we aim to minimize the mean absolute error[6] between the true count and the estimated count. As in the nearest neighbor setup, the two constraints of interest are the size of the data structure and the number of lookups for query execution. Consequently, our framework can be easily adapted to model this problem. We also choose this problem to highlight the versatility of our framework as it can be applied to streaming settings.

**Data processing Network**   We model the data structure as a $k$ dimensional vector $\hat{D}$ and use an MLP as the data-processing network which is responsible for writing to $\hat{D}$. When a new element arrives in our stream, we allow the model to update M values in the data structure. Specifically, when an element arrives at time-step $t$, the data-processing network outputs $M$ $k$-dimensional update position vectors $u_1, ..., u_M$ and M corresponding scalar update values $v_1, ..., v_M$. We then apply the update, obtaining $\hat{D}_{t+1} = \hat{D}_t + \sum_{i=1}^{M} u_i * v_i$. Unlike in the NN setting where we did not constrain the construction complexity of the data structure, here we have limited each update to the data structure to a budget of $M$ lookups. We do so as in the streaming setting it is assumed updates occur often and so it is less reasonable to consider them as a one-time construction overhead cost.

**Query processing Network**   Query processing is handled in a similar fashion to NN search — we have $M$ query MLP models that output lookup positions. Finally, we also train a predictor network $\psi(v_1, ..., v_M)$[7] that takes in the $M$ values retrieved from the lookups and outputs the final prediction.

---

[6]We use absolute error as this is the metric commonly used in prior work (Cormode & Muthukrishnan, 2005; Cormode & Hadjieleftheriou, 2010) but our setup works for squared error as well.

[7]We use the same MLP architecture for $\psi$ as we use for the query-models.

EXPERIMENTS

**Zipfian Distribution**   We evaluate our model in a setting where both the stream and query distributions follow a Zipfian distribution. This simulates a common feature of frequency-estimation datasets where a few "heavy hitter" elements are updated or queried more frequently than others (Hsu et al., 2019). For each training instance, the rank order of the elements in the domain is consistent across both the stream and query distributions, but it is randomized across different training instances. As a result, the model cannot rely on knowing which specific elements are more frequent than others; only the overall Zipfian skew is consistent across training instances.

We use a data structure of size $k = 32$ and train our model with $M \in \{1, 2, 4\}$ queries. Both the data and query distributions are Zipfian over $\{1, ..., 1000\}$ with a fixed skew of $\alpha = 1.2$. We evaluate the mean absolute error over streams of length 100 and compare with the CountMinSketch algorithm, a hashing-based method for frequency estimation (Cormode & Muthukrishnan, 2005) (See App. E for an overview). Our model's performance improves with more queries and outperforms CountMinSketch (Figure 7). In this case, CountMinSketch degrades with more queries as for a fixed size memory ($k = 32$), it is more effective for this distribution to apply a single hash function over the whole memory than to split the memory into $k$ partitions of size $k/M$ and use separate hash functions. We look at the learned algorithm in more detail and find that our model learns an algorithm similar to CountMinSketch, but with an important difference: it uses an update delta of less than 1 when a new item arrives, instead of the delta of 1 used by CountMinSketch. We find that this can be particularly useful when the size of the data structure is small and collisions are frequent. We hypothesize that the better performance of the learned solution is at least partially due to the smaller delta (Figure 21).

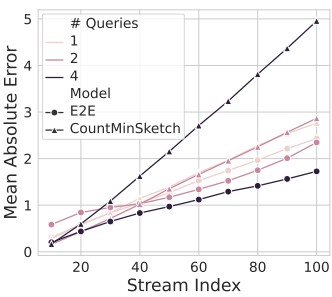

Figure 7:   When estimating frequencies of elements drawn from a randomly ordered Zipfian distribution, our model outperforms the CountMinSketch baseline given 1, 2, and 4 queries.

**Learning Heavy Hitter Features**   In the previous experiment, the Zipfian distribution shape was fixed across training instances but the rank ordering of elements was random. In some settings, however, it may be possible to predict which elements are more likely to occur in the stream. While the exact elements may vary between streams, frequently occurring items could share features across streams. For instance, Hsu et al. (2019) show that in frequency estimation for network flows, certain types of IP addresses receive much more traffic than others. We simulate this setting by fixing the rank ordering of the Zipfian distribution. However, instead of using a universe of integer elements $\{1, ..., K\}$, we instead use their corresponding 3-digit MNIST images with $K = 100$ (constructed as in the MNIST NN experiment). Given a stream of integers, we map them to their corresponding MNIST labels and then for each label we sample a corresponding image from the training set. During evaluation, we use images samples from the test set. As the distribution is skewed and the ranking is fixed, images with smaller numbers are sampled much more frequently than those with larger numbers. As in the MNIST NN experiment, we also use a feature-learning CNN model to process the images before passing them to the data-processing and query-execution networks.

We compare our model to CountMinSketch with 1-query that is given the underlying labels instead of the images. Our model has a significantly lower error than the baseline (0.15 vs 2.81 averaged over a stream of size 100 (see Fig. 22)) as the latter is distribution-independent. By training from the data-distribution end-to-end, our framework is able to simultaneously learn features of heavy hitters (in this case, clustering images with the same label) and use this information to design an efficient frequency estimation data structure. We investigate the learned structure and find that the model has reserved separate memory positions for heavy hitters, thereby preventing collisions (Fig. 23).

## 3.2   OTHER POTENTIAL APPLICATIONS

Here, we outline several other potential applications of our framework to facilitate future work.

**Graph data structures**: Many graph-related problems require an efficient representation to support connectivity or distance queries between vertices. For distance queries, one approach is to use quadratic space to store the distances between all vertex pairs, allowing $O(1)$ query time. Alter-

natively, one could use no extra space and simply store the graph (which may require significantly less than quadratic space) and run a shortest-path algorithm at query time. The challenge is to find a middle ground: using sub-quadratic space while still answering distance queries faster than a full shortest-path computation (Thorup & Zwick, 2005).

**Sparse matrices:** Another common problem that can be framed as a data structure problem is that of compressing sparse matrices. Given an $M \times N$ matrix, on one hand, one could store the full matrix and access elements in $O(1)$ time. However, depending on the number and distribution of 0s in the matrix, different data structures could be designed that use less than $O(MN)$ space. There is an inherent trade-off between how compressed the representation is and the time required to access elements of the matrix to solve various linear algebraic tasks involving the matrix such as matrix-vector multiplication (Buluç et al., 2011; Chakraborty et al., 2018).

**Learning statistical models:** Our framework can also handle problems such as learning statistical models like decision trees, where the input to the data-processing network is a training dataset, and the output is a model such as a decision tree. The query algorithm would then access a subset of the model at inference time, such as by doing a traversal on the nodes of the decision tree. This could be used to explore questions around optimal algorithms and heuristics for learning decision tress, which are not properly understood (Blanc et al., 2021; 2022).

## 4 RELATED WORK

**Learning-Augmented Algorithms** Recent work has shown that traditional data structures and algorithms can be made more efficient by learning properties of the underlying data distribution. Kraska et al. (2018) introduced the concept of learned index structures which use ML models to replace traditional index structures in databases, resulting in significant performance improvements for certain query workloads. By learning the cumulative distribution function of the data distribution the model has a stronger prior over where to start the search for a record, which can lead to provable improvements to the query time over non-learned structures (Zeighami & Shahabi, 2023). Other works augment the data structure with predictions instead of the query algorithm. For example, Lin et al. (2022a) use learned frequency estimation oracles to estimate the priority in which elements should be stored in a treap. Perhaps more relevant to the theme of our work is Dong et al. (2019), which trains neural networks to learn a partitioning of the space for efficient nearest neighbor search using locality sensitive hashing, and the body of work on learned hash functions (Wang et al., 2015; Sabek et al., 2022). While all these works focus on augmenting data structure design with learning, we explore whether data structures can be discovered entirely end-to-end using deep learning.

**Neural Algorithmic Learners** There is a significant body of work on encoding algorithms into deep networks. Graves et al. (2014) introduced the Neural Turing Machine (NTM), which uses external memory to learn tasks like sorting and copying. Veličković et al. (2019) used graph neural networks (GNNs) to encode classical algorithms such as breadth-first search. These works train deep networks with a great degree of supervision with the aim of encoding known algorithms. For instance, Graves et al. (2014) use the ground truth sorted list as supervision to train the model to sort. There has also been work on learning algorithms in an end-to-end fashion. Fawzi et al. (2022) train a model using reinforcement learning to discover matrix multiplication algorithms, while Selsam et al. (2018) train neural networks to solve SAT problems. Garg et al. (2022) show that transformers can be trained to encode learning algorithms for function classes such as linear functions and decision trees. Our work adds to this line of research on E2E-learning, focusing on discovering data structures.

## 5 CONCLUSION

We began with the question of whether deep learning models can be trained to discover data structures from scratch. This work provides initial evidence that it is possible. For both nearest neighbor search and frequency estimation, the models—trained end-to-end—discover distribution-dependent data structures that outperform worst-case baselines. We hope this research inspires further exploration into data structure and algorithm discovery.

One limitation that future research could address is scale. Due to computational constraints, most of our experiments are conducted with datasets of size $N = 100$, although in App. F we scale to $N = 500$. While this is reasonable for gaining insights into data structure design, practical end-to-end use would likely require further scaling. We believe both larger models and better inductive biases could enable scaling up further (see App. F for details).

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

# APPENDIX

## A  TRAINING DETAILS

### A.1  NEAREST NEIGHBORS

The transformer in the data-processing network is based on the NanoGPT architecture (Karpathy, 2024) and has 8 layers with 8 heads each and an embedding size of 64. Each query model $Q_\theta^i$ is a 3-layer MLP with a hidden dimension of size 1024. Each hidden layer consists of a linear mapping followed by LayerNorm (Ba et al., 2016) and the ReLU activation function (Nair & Hinton, 2010). In all experiments we use a batch size of 1024, 1e-3 weight decay and the Adam optimizer (Kingma & Ba, 2017) with default PyTorch (Paszke et al., 2017) settings. We do a grid search over $\{0.0001, 0.00001, 0.00005\}$ to find the best learning rate for both models. All models are trained for at most 4 million gradient steps with early-stopping. We apply the Gumbel Softmax (Jang et al., 2017) with a temperature of 2 to the lookup vectors to encourage sparsity. All experiments are run on a single NVIDIA RTX8000 GPU.

### A.2  FREQUENCY ESTIMATION

We follow the same setup as the nearest neighbors training except for frequency estimation, the data-processing network is a 3-layer MLP with a hidden dimension of size 1024. We do a grid search over $\{0.0001, 0.00005, 0.00001\}$ to find the best learning rate for both models. Models are trained for 200k gradient steps with early stopping. All experiments are run on a single NVIDIA RTX8000 GPU.

## B  ADDITIONAL NEAREST NEIGHBOR PERFORMANCE PLOTS

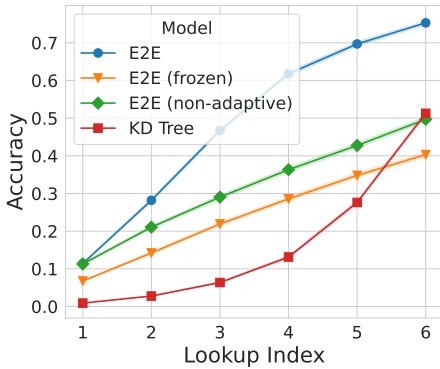

Figure 8: 2D Uniform Accuracy.

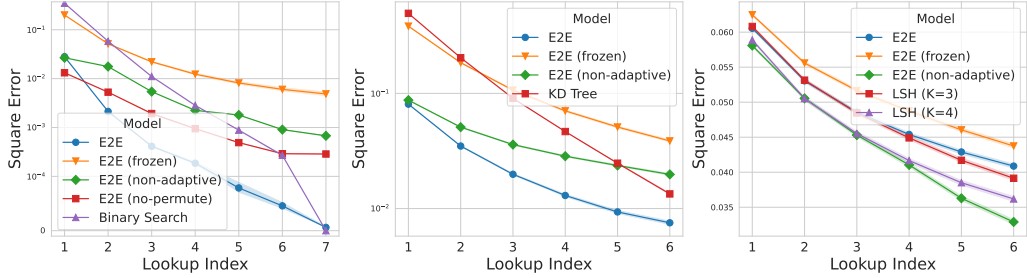

Figure 9: Mean square error plots for **(Left)** 1D Uniform distribution, **(Center)** 2D Uniform distribution, **(Right)** 30D Uniform distribution over unit hyper-sphere.

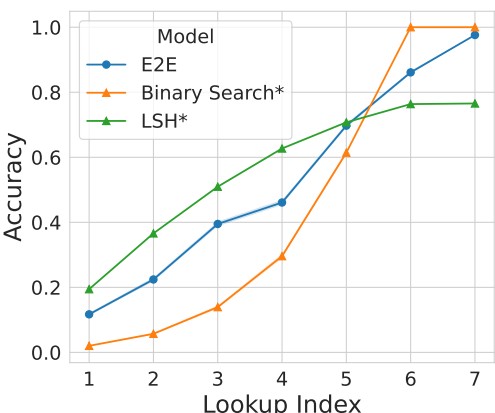

Figure 10: 3-Digit MNIST Nearest Neighbors Accuracy. Even though binary search (over the underlying digits) is an unfair comparison, we include it as a reference to compare our model's performance with.

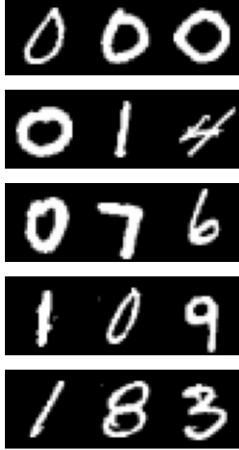

Figure 11: Samples from 3-Digit MNIST

### B.1 HARD DISTRIBUTION

To generate data from the hard distribution, we first sample the element at the 50th percentile from the uniform distribution over a large range. We then sample the 25th and 75th percentile elements from a smaller range and so on. The intuition behind this distribution is to reduce concentration such that $p(NN|q)$ is roughly uniform where $NN$ denotes the index of the nearest-neighbor of $q$ in the sorted list.

Precisely, to sample $N$ points from the hard distribution we generate a random balanced binary tree of size $N$. All vertices are random variables of the form $Uniform(0, a^{\log n - k})$ where $a$ is some constant and $k$ is the level in the tree that the vertice belongs to. If the $i - th$ node in the tree is the left-child of its parent, we generate the point $x_i$ as $x_i = x_{p(i)} - d_i$ where $p(i)$ denotes the parent of the $i - th$ node and $d_i$ is a sample from node $i$ of the random binary tree. Similarly, if node $i$ is the right child of its parent, $x_i = x_{p(i)} + d_i$. For the root element $x_0 = d_0$. In our experiments we set $a = 7$. The larger the value of $a$, the greater the degree of anti-concentration. We found it challenging to train models with $N > 16$ as the range of values that $x_i$ can take increases with $N$. Thus for larger $N$, the model needs to deal with numbers at several scales, making learning challenging.

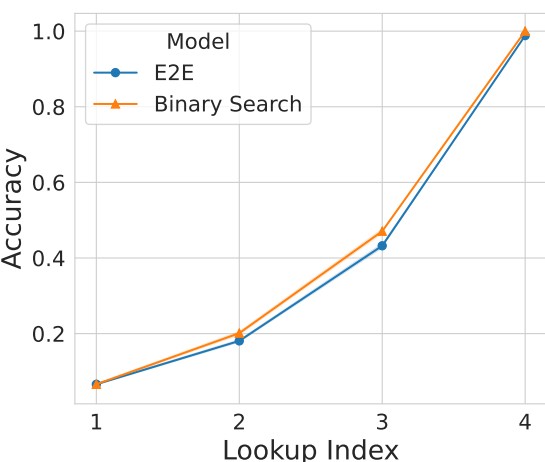

Figure 12: Our model's performance is closely aligned with binary search on the hard distribution in 1D. By design, this distribution does not have a useful prior our model can exploit and so it learns a binary search like solution.

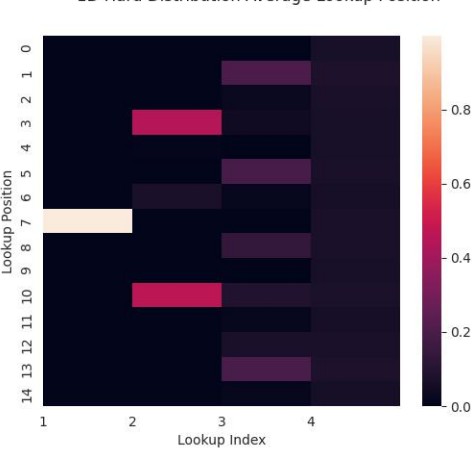

Figure 13: The positional distribution per lookup in the 1D Hard experiment. Our model closely aligns with binary search, first looking at the middle element, then (approximately) either the 25th or 75th percentile elements, and so on.

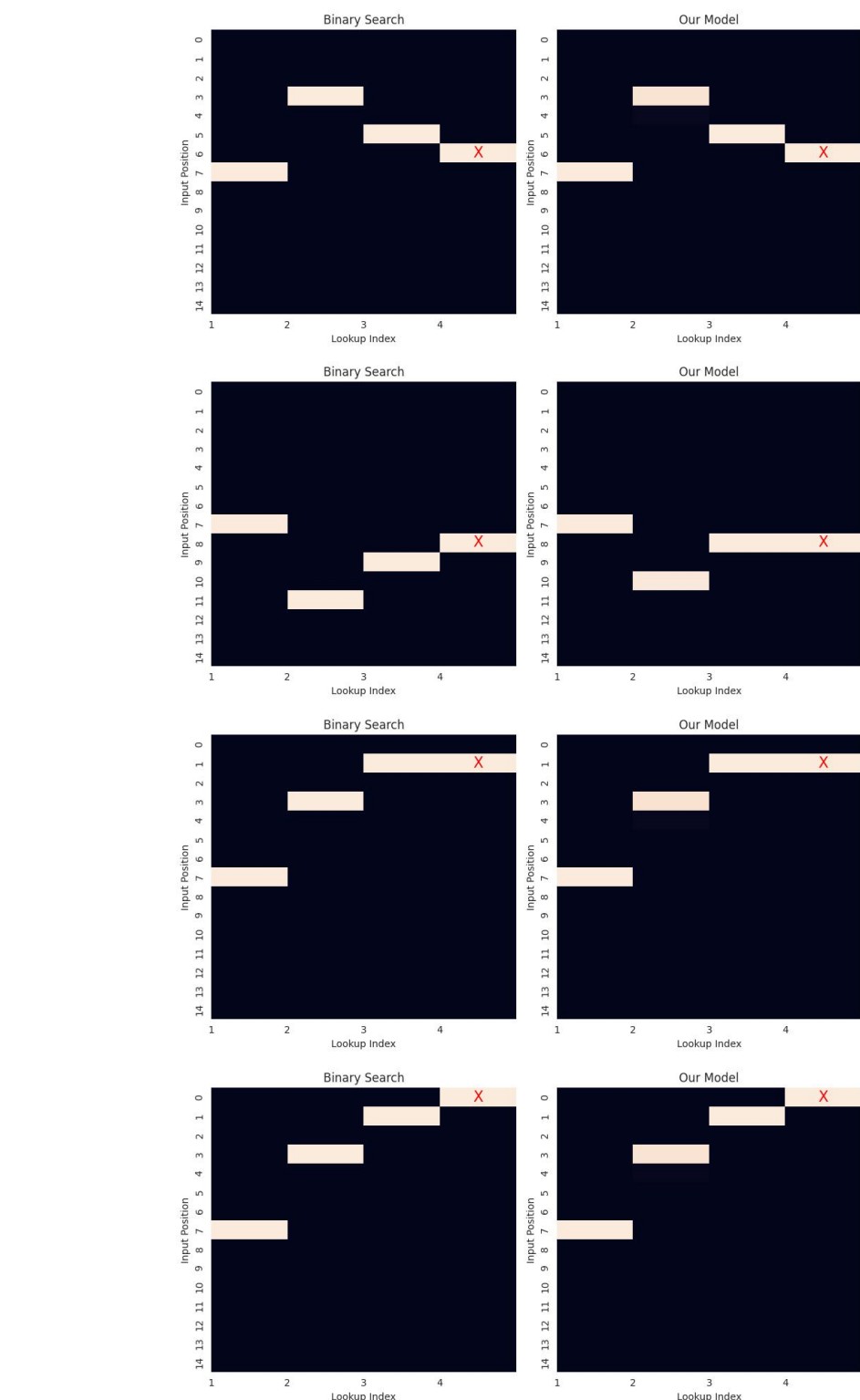

Figure 14: Binary Search vs. our model on the hard distribution in 1D.

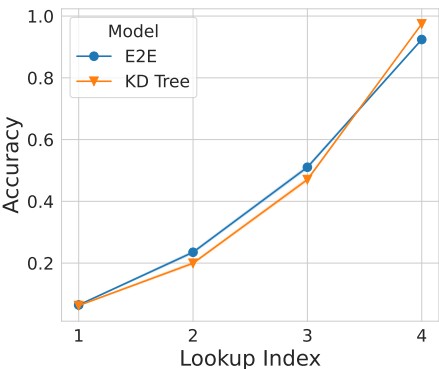

Figure 15: On the 2D hard distribution our model roughly tracks the performance of a k-d tree.

## C    ADDITIONAL EXPERIMENT FINDINGS

### C.1    NOISE INJECTION FOR LOOKUP SPARSITY

We find that adding noise prior to applying the soft-max on the lookup vector $m_i$ leads to sparser queries. We hypothesize that this is because the noise injection forces the model to learn a noise-robust solution which corresponds to a sparse solution. Consider a simplified setup in 1D where the query model is not conditioned on $q$ and is only allowed one lookup ($M = 1$) and $D$ is a sorted list of three elements: $D = [x_1, x_2, x_3]$. For a given query $q$ and its nearest neighbor $y$, the query-execution network is trying to find the optimal vector $\hat{m} \in \mathbb{R}^3$ that minimizes $||y - m^T D||_2^2$ where $m = softmax(\hat{m} + \epsilon), \epsilon \sim$ Gumbel distribution Jang et al. (2017). Given that $M = 1$, the model cannot always make enough queries to identify $y$ and so in the absence of noise the model may try to predict the 'middle' element by setting $\hat{m}_1 = \hat{m}_2 = \hat{m}_3$. However, when noise is added to the logits $\hat{m}$ this solution is destabilized. Instead, in the presence of noise, the model can robustly select the middle element by making $\hat{m}_2$ much greater than $\hat{m}_1, \hat{m}_3$. We test this intuition by running this experiment for large values of $N$ and find that with noise the average gradient is much larger for $\hat{m}_{N/2}$.

### C.2    2D UNIFORM DISTRIBUTION

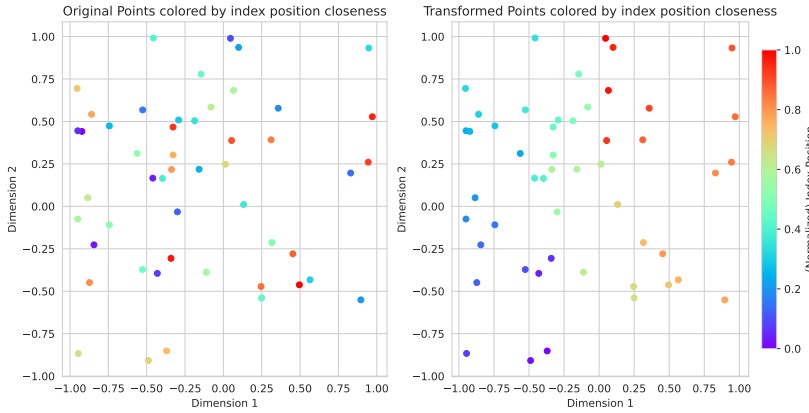

Figure 16: Our model's learned permutation on the 2D uniform distribution. The model puts elements that are close together in the Euclidean plane next to each other in the permutation.

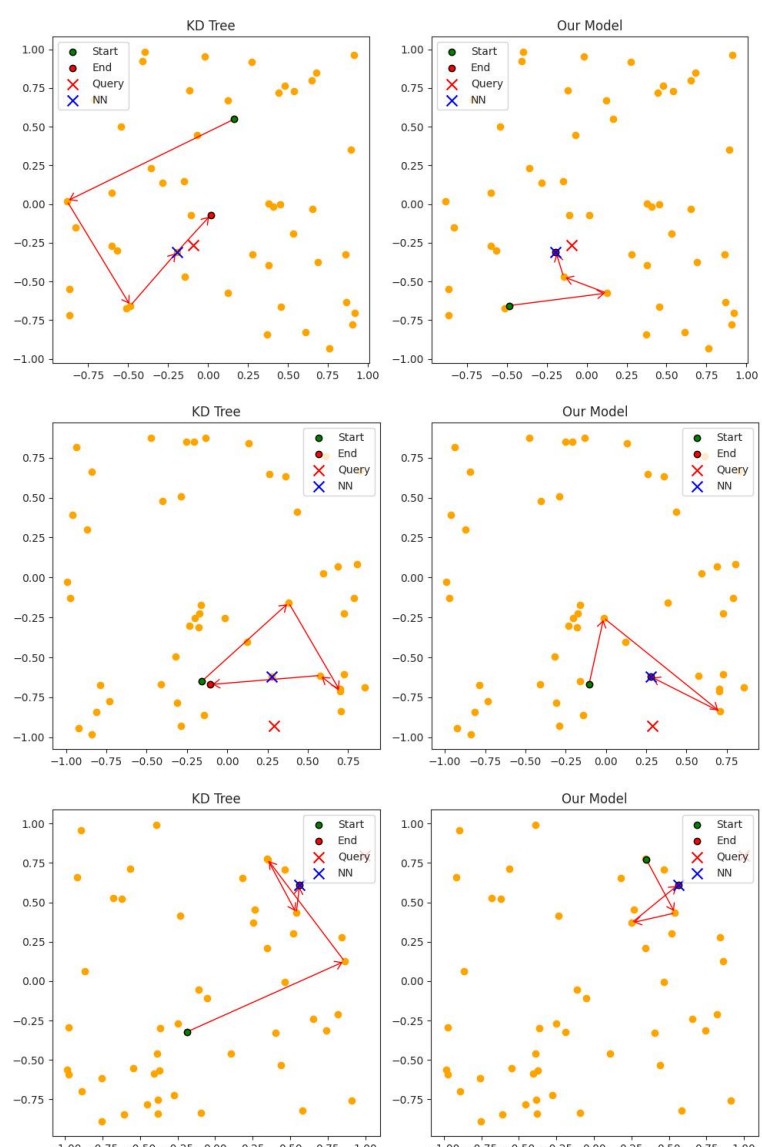

Figure 17: k-d search vs. our model on the uniform distribution in 2D. Unlike the k-d tree, our model has a stronger prior over where to begin its search.

## C.3    N=8, M=1 30D EXPERIMENT

To determine if our model has learned an LSH-like solution, we try to reverse engineer the query model in a simple setting where $N = 8$ and $M = 1$. The query-execution model is only allowed one lookup. We fit 8 one-vs-rest logistic regression classifiers using queries sampled from the query distribution and the output of the query model (lookup position) as features and labels, respectively. We then do PCA on the set of 8 classifier coefficients. We find that the top 2 principal components explain all of the variance which suggests that the query model's mapping can be explained by the projection onto these two components. In Figure 18 we plot the projection of queries onto these components and color them based on the position they were assigned by the query model. We do the same for inputs $x_i \in D$ and color them by the position they were permuted to. The plot on the right suggests that the data-processing network permutes the input vectors based on their projection onto these two components. This assignment is noisy because there may be multiple inputs in a dataset that map to the same bucket and because the model can only store a permutation, some buckets

experience overflow. Similarly, the query model does a lookup in the position that corresponds to the query vector's bucket. This behaviour suggests the model has learned a locality-sensitive hashing type solution!

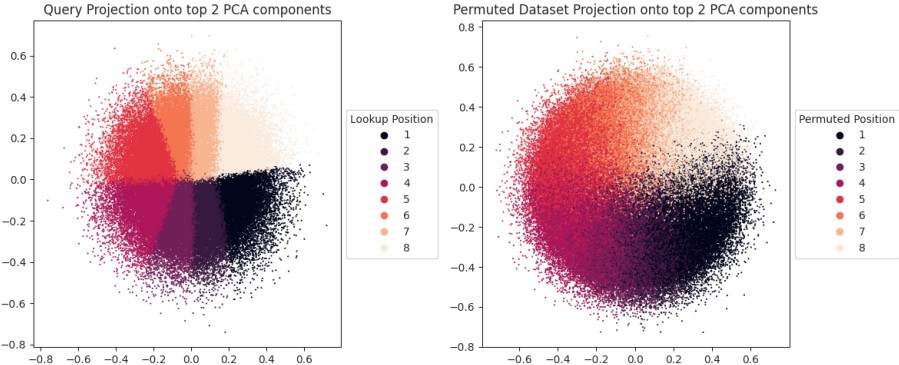

Figure 18: **(Left)** Projection of queries onto top two PCA components of the decision boundaries of the query model, colored by the lookup position the query is mapped to. **(Right)** Projection of inputs onto the same PCA components colored by the position the data-processing model places them in. Both the data-processing and query models map similar regions to the same positions, suggesting an LSH-like bucketing solution has been learned.

## C.4   1D EXTRA SPACE

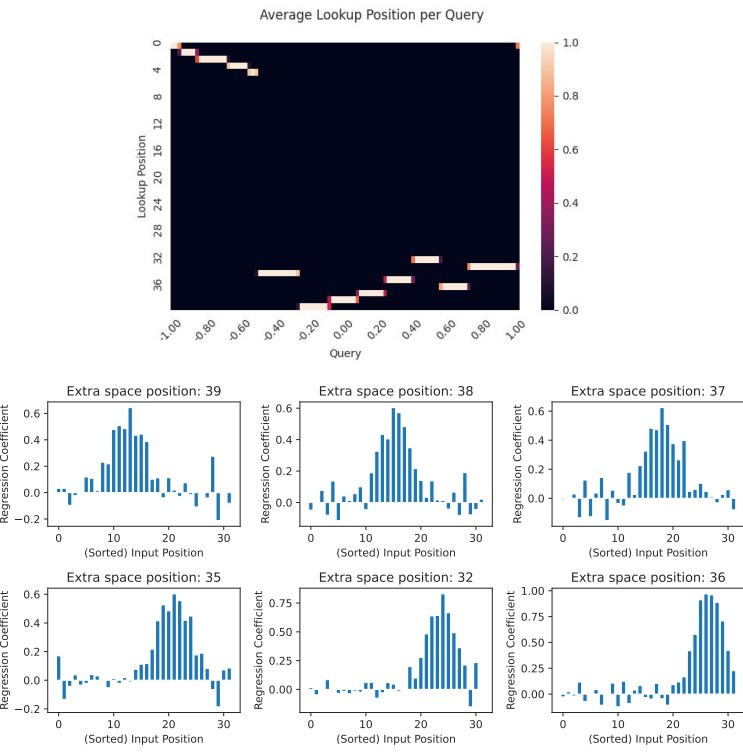

Figure 19: **(Top)** Decision boundary of the first query model. **(Bottom)** The regression coefficients of the values stored in extra positions as a linear function of the (sorted) inputs.

### C.4.1 BUCKET BASELINE

We create a simple bucket baseline that partitions $[-1, 1]$ into $T$ evenly sized buckets. In each bucket $b_i$ we store $argmin_{x_j \in D}||x_j - l_i||$ where $l_i$ is the midpoint of the segment partitioned in $b_i$. This baseline maps a query to its corresponding bucket and predicts the input stored in that bucket as the nearest-neighbor. As $T \to \infty$ this becomes an optimal hashing-like solution.

### C.4.2 UNDERSTANDING EXTRA SPACE USAGE

By analyzing the lookup patterns of the first query model, we can better understand how the model uses extra space. In Figure 19 we plot the decision boundary of the first query model. The plot demonstrates that the model chunks the query space ($[-1, 1]$) into different buckets. To get a sense of what the model stores in the extra space, we fit a linear function on the sorted inputs and regress the values stored in each of the extra space tokens $b_i$ and plot the coefficients for several of the extra spaces in Figure 19. For a given subset of the query range, the value stored at its corresponding extra space is approximately a weighted sum of the values stored at the indices that correspond to the percentile of that query range subset. This is useful information as it tells the model for a given query percentile how 'shifted' the values in the current dataset stored in the corresponding indices are from model's prior.

## C.5 30D EXTRA SPACE

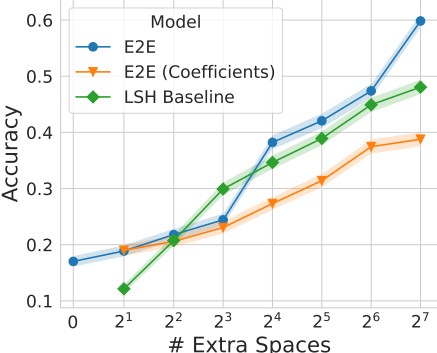

Figure 20: Our unconstrained model (E2E) and a more interpretable version (E2E (Coefficients)) both learn to effectively leverage an increasing amount of extra space in 30D, with the unconstrained model outperforming an LSH baseline.

In high-dimensions it is less clear what solutions there are to effectively leverage extra space, and in fact understanding optimal tradeoffs in this case is open theoretically (Arya et al., 1998).

We follow a similar setup to the 1D extra space experiments but use the data and query distributions from section 2.2. We experiment with two versions of extra space (unrestricted) and (coefficients). For the unrestricted version the data model can store whatever 30 dimensional vector it chooses in each of the extra spaces. For the coefficient model, instead of outputting a 30 dimensional vector, for each extra space, the model outputs a separate N dimensional vector of coefficients. We then take a linear combination of the (permuted) input dataset using these coefficients and store the resulting vector in the corresponding extra positions. While the unrestricted version is more expressive the coefficient version is more interpretable. We include both versions to demonstrate the versatility of our framework. If one is only interested in identifying a strong lower-bound of how well one can use a fixed budget of extra space they may use the unrestricted model. However, if they are more concerned with investigating specific classes of solutions or would like a greater degree of interpretability they can easily augment the model with additional inductive biases such as linear coefficients.

We plot the performance of both models along with an LSH baseline in Figure 20. While both models perform competitively with an LSH baseline and can effectively leverage an increasing amount of extra space, the unrestricted model outperforms the coefficient model at a certain point.

## C.6 FREQUENCY ESTIMATION

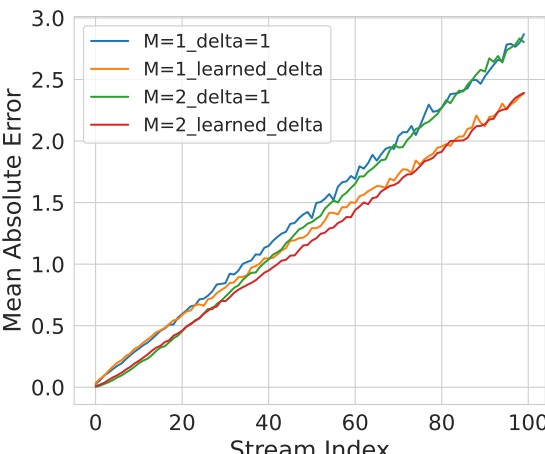

Figure 21: We apply the insight that we learned from our E2E model to improve CountMinSketch on the Zipfian distribution. By changing the CountMinSketch update delta of 1 to our model's learned delta ($\Delta = 0.87$ for $M = 1$ and $\Delta = 0.93$ for $M = 2$), we can improve the performance of CountMinSketch on $M \in \{1, 2\}$ queries.

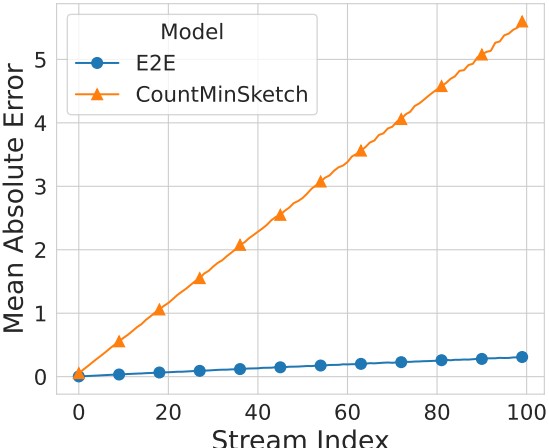

Figure 22: On the MNIST heavy-hitters frequency estimation experiment, our model significantly outperforms CountMinSketch. This is because our model can learn features predictive of heavy hitters, as opposed to the distribution-agnostic CountMinSketch.

## D LSH BASELINE

Our LSH baseline samples $K$ random vectors $\mathbf{r_1}, ..., \mathbf{r_K}$ from the standard normal distribution in $\mathbb{R}^d$. For a given vector $\mathbf{v} \in \mathbb{R}^d$, its hash code is computed as $hash(\mathbf{v}) = [sign(\mathbf{v^T r_1}), ..., sign(\mathbf{v^T r_K})]$. In total, there are $2^K$ possible hash codes. To create a hash table, we assign each hash code a bucket of size $N/2^K$. For a given dataset $D = \{x_1, ..., x_N\}$, we place each input in its corresponding

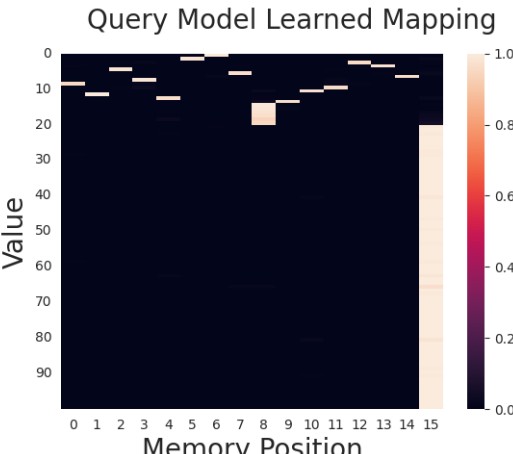

Figure 23: We show the decision boundary learned by the query/data-processing network in the MNIST heavy hitters experiment. As images with smaller numbers occur more frequently in the stream, the memory-constrained model learns to reserve separate memory positions for these items in order to prevent collisions among them.

bucket (determined by its hash code $hash(x_i)$. If the bucket is full, we place $x_i$ in a vacant bucket chosen at random. Given a query $q$ and a budget of $M$ lookups, the baseline retrieves the first $M$ vectors in the bucket corresponding to $hash(q)$. If there are less than $M$ vectors in the bucket, we choose the remaining vectors at random from other buckets. We design this setup like so to closely align with the constraints of our model (i.e. only learning a permutation).

## E CountMinSketch

CountMinSketch (Cormode & Muthukrishnan, 2005) is a probabilistic data structure used for estimating the frequency of items in a data stream with sublinear space. It uses a two-dimensional array of counters and multiple independent hash functions to map each item to several buckets. When a new item $x$ arrives, the algorithm computes $d$ hash functions $h_1(x), h_2(x), \ldots, h_d(x)$, each of which maps the item to one of $w$ buckets in different rows of the array. The counters in the corresponding buckets are incremented by 1. To estimate the frequency of an item $x$, the minimum value across all counters $C[1, h_1(x)], C[2, h_2(x)], \ldots, C[d, h_d(x)]$ is returned. The sketch guarantees that the estimated frequency $\hat{f}(x)$ of an item $x$ is at least its true frequency $f(x)$, and at most $f(x) + \epsilon N$, where $N$ is the total number of items processed, $\epsilon = \frac{1}{w}$, and $w$ is the width of the sketch. The probability that the estimate exceeds this bound is at most $\delta = \frac{1}{d}$, where $d$ is the depth of the sketch (i.e., the number of hash functions). These guarantees hold even in the presence of hash collisions, providing strong worst-case accuracy with $\mathcal{O}(w \cdot d)$ space.

## F Limitations and Future Work

One limitation of our work is the scale at which we learn data structures. Most of our nearest neighbor search experiments are done with input dataset sizes around $N = 100$, however, we are also able to scale up to $N = 500$ (Figure 24 (Left/Center)), though with less than $\log(N)$ queries. While we demonstrate that useful data structures can still be learned at this scale, it is possible that other classes of structures only emerge for larger datasets. We also believe that many of the insights that can be derived from our models' learned solutions would scale to larger $N$. For instance, sorting in 1D and locality-sensitive hashing in higher dimensions. We limit ourselves to datasets of these sizes due to computational constraints, and because our primary goal was to understand whether end-to-end data structure design is feasible at any reasonable scale. However, we believe our framework could scale to datasets with thousands of points by increasing the parameter counts of the data-processing and query-execution models. Moreover, as transformers become increasingly

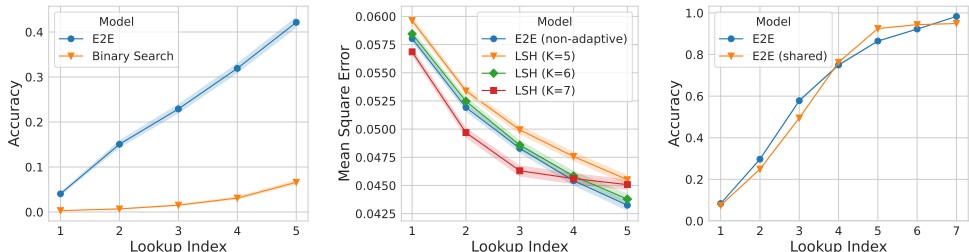

Figure 24: We scale both the 1D (**Left**) and 30D (**Center**) experiments to datasets of size $N = 500$. (**Right**) We compare our E2E model with a version where the query-execution network is only composed of one query-model (E2E (shared)) that is used in a loop for $M = 7$ queries during training on the 1D Uniform distribution, thereby conserving parameters by reusing weights. This could be a promising direction for problem settings where there is a recursive structure to the query algorithm.

efficient at handling larger context sizes in language modeling settings, some of these modeling advancements may also be used for scaling models in the context of data structure discovery.

Complementary to our work, it could also be valuable to explore better inductive biases for the query and data-processing networks, and other methods to ensure sparse lookups, enabling smaller models to scale to larger datasets. For instance, using shared weights among query models can be helpful in scaling up the number of queries. As a first step in this direction we show that a single query model can be used in-a-loop for NN search in 1D (Figure 24 (Right)). However, we leave further investigation for future work.

# G  SUPPLEMENTARY EXPERIMENTS

## G.1  IMPROVED FREQUENCY ESTIMATION WITH AUGMENTED COUNTMINSKETCH

In our experiments on learning frequency estimation algorithms (Section 3.1), we found that on the Zipfian distribution our model was able to outperform the CountMinSketch algorithm by using a smaller update delta. We use this insight to design a modified version of CountMinSketch that uses a custom update delta. In Figure 25, we show that this augmented CountMinSketch algorithm can outperform vanilla CountMinSketch on the large-scale CAIDA IP traffic dataset (CAIDA, 2016) by up to a factor of two. These results demonstrate that even at small scale, our model can provide useful insight into data structure design that can be transferred to realistic settings.

The CAIDA dataset (CAIDA, 2016) consists of traffic data collected in 2016 from a backbone link of a Tier-1 ISP between Chicago and Seattle. Each recording session spans approximately one hour, capturing around 30 million packets and 1 million unique flows per minute. We use the first minute for our experiment.

## G.2  ADDITIONAL HIGH-DIMENSIONAL NN EXPERIMENTS WITH REALISTIC DATA

We run additional nearest-neighbor experiments on two standard high-dimensional approximate nearest-neighbor benchmarks: SIFT and FashionMNIST (projected to 100 dimensions via PCA) (Aumüller et al., 2020) to further demonstrate our model can handle more realistic data. In addition to locality-sensitive hashing, we include comparisons to several learning-to-hash baselines: ITQ, K-Means, and NeuralLSH. See (Dong et al., 2019) for more details about these baselines. Our model performs competitively with these learning-to-hash baselines (Figure 26). We do not expect our model to outperform these baselines in this setting as it is unclear that query adaptivity should be beneficial here. Rather, we include these results to further emphasize that our end-to-end model can recover reasonable solutions in a variety of settings - even when compared to carefully hand-designed solutions. We also show an example of how the data-processing model learns to transform the FashionMNIST data by organizing the images by class (Figure 27).

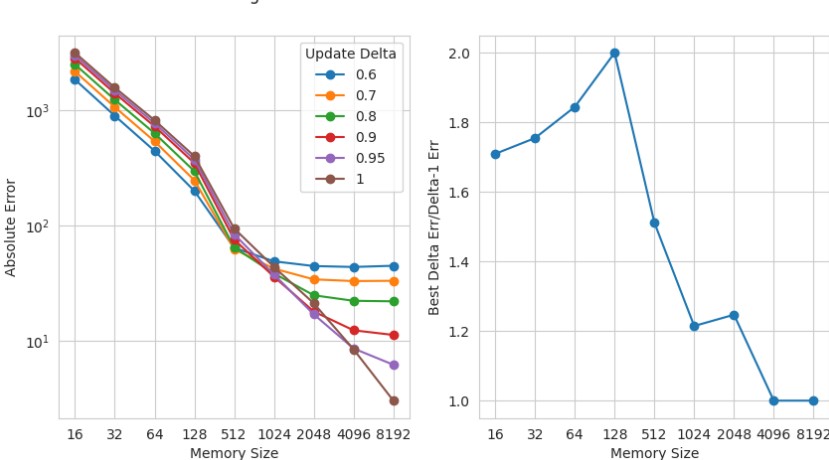

Figure 25: (**Left**) CountMinSketch performance on the CAIDA dataset with different update deltas vs. memory size. (**Right**) The relative performance of the best update delta vs the default delta ($\Delta = 1$) for different memory sizes. In some regimes, our augmented CountMinSketch can perform up to twice as well as vanilla CountMinSketch just by modifying the update delta.

For each dataset, we use the train split to train our model (as well as the learning-to-hash baselines) and for evaluation we use the test split. The learning-to-hash baselines cluster the data into 16 partitions and we then use the same setup described in Appendix D to execute the NN search. We chose 16 partitions as this produces the best performance for the learning-to-hash baselines in our setup where $N = 100$ and $M = 6$.

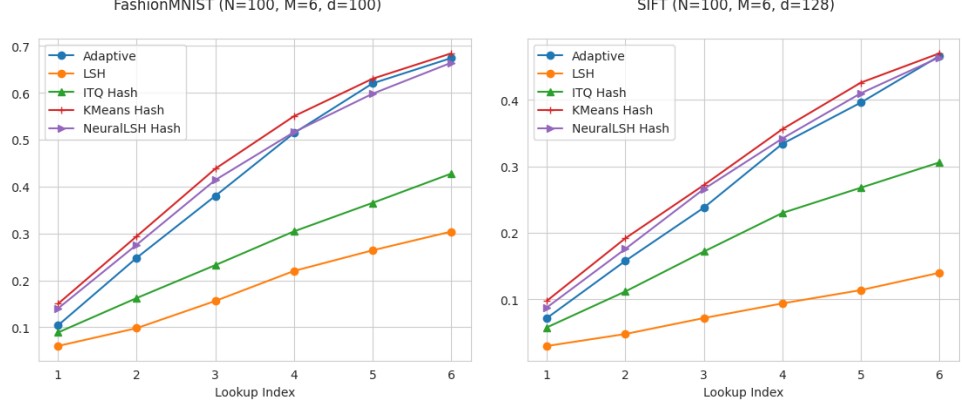

Figure 26: Our model compared to various hashing baselines on the FashionMNIST dataset (**left**) and the SIFT dataset (**right**).

## G.3 NN EXPERIMENTS WITH LINEAR TRANSFORMER

To demonstrate that in some settings quadratic attention can be substituted with a cheaper alternative, we run additional experiments in 2D (uniform distribution) and in high dimensions (Fashion-MNIST) with the linear attention Performer model (Choromanski et al., 2020)[8]. We use the same model hyper-parameters as the quadratic attention model and plot the results in Figure 28. The comparable performance of the linear attention model suggests that it could be a computationally-cheaper alternative that can enable scaling up models to larger settings.

---

[8] We use the Pytorch implementation from https://github.com/lucidrains/performer-pytorch/tree/main.

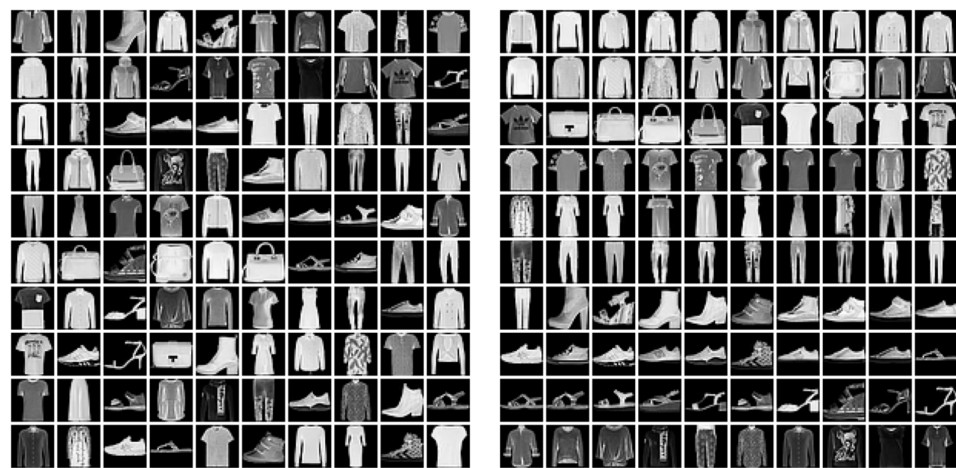

Figure 27: (**Left**) Sample raw dataset from FashionMNIST (**Right**) The learned data structure. The data-processing model learns to cluster similar items together.

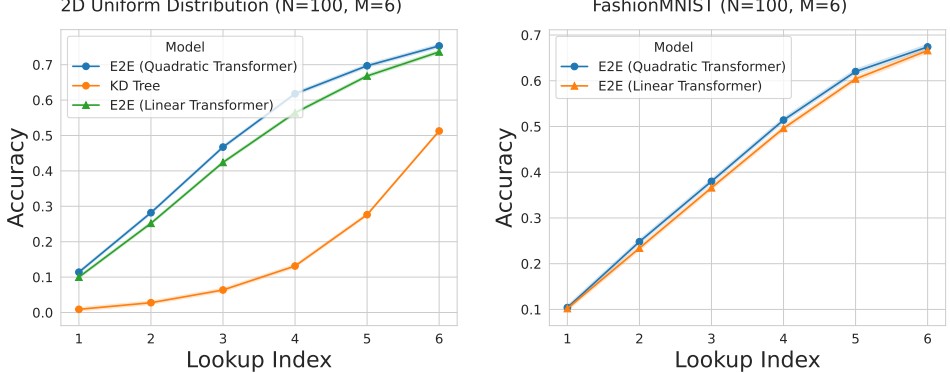

Figure 28: Performance of the quadratic attention transformer vs. linear attention performer transformer (Choromanski et al., 2020) in both 2D (**left**) and high dimensions (100) (**right**).

