# OpenReview forum: "Discovering Data Structures: Nearest Neighbor Search and Beyond"
_ICLR.cc/2025/Conference — Submitted to ICLR 2025_

### Official Review · Reviewer_mekY · 2024-10-31

**Soundness:** 3
**Presentation:** 2
**Contribution:** 2
**Rating:** 5
**Confidence:** 4

**Summary:**

This paper presents a framework for end-to-end learning of data structures, enabling deep learning models to autonomously discover efficient data structures optimized for specific data distributions. The framework is applied to two problems: nearest neighbor (NN) search and frequency estimation. Key findings include the model learning sorting and search strategies for NN search that rival or outperform traditional methods like binary search, k-d trees, and locality-sensitive hashing (LSH) in some cases. Beyond NN search, the authors apply the model to frequency estimation in data streams, where it demonstrates competitive performance with traditional data structures like CountMin Sketch.

**Strengths:**

S1. The paper is well-motivated and the problem is interesting. The paper is motivated based on the observation that many data structures are designed to be worst-case optimal and agnostic to the underlying data and query distribution. Neural networks show potential in other tasks e.g. SAT solving. So it is natural to study if neural networks can learn data structures that are optimal for specific data and query distribution in an end-to-end manner.

S2. Experiments show that the proposed framework shows a potential to learn the distribution-specific data structures that are more efficient than the traditional worst-case optimal data structures with synthetic data and the mnist dataset, for the problem of NN search and frequency estimation. This demonstrates the potential of applying neural networks to learn data structures.

**Weaknesses:**

W1: The paper’s reliance on a transformer model with self-attention for data processing presents efficiency and scalability issues for the NN search task. The self-attention mechanism incurs a computational complexity of $O(n^2)$, limiting its application to datasets with a few hundred points due to memory and processing constraints. Considering the NN search problem has a trivial solution with $O(n)$ complexity, this constraint raises concerns about the framework's practicality in real-world applications where much larger datasets are typical. A more scalable approach, such as a different backbone structure, could significantly extend the applicability of this work by reducing computational demands and enabling a larger $n$. Without such efficiency improvements, the method is impractical.

W2: Most experiments rely on synthetic data, except for the MNIST image dataset, which limits the framework's demonstration of generalizability to learn challenging real-world data distributions. While synthetic data offers control over distributions, testing on real-world datasets with unknown or complex distributions would better illustrate the framework’s ability to discover meaningful data structures.

W3: The paper’s methodology section lacks sufficient clarity on some technical aspects.
- (1) The paper uses NanoGPT, a decoder-only transformer that typically requires discrete tokens for input and output. However, the paper doesn’t clearly explain how continuous data inputs are tokenized or how outputs are transformed into the scalar ranking $o_i$ values. More detail on the embedding and decoding process for continuous data is necessary to understand the data flow and model structure.
- (2) In experiments involving CNNs, essential details about the CNN architecture, such as the number of layers, kernel sizes, or feature extraction strategies, are missing. Including these specifications would make it easier to interpret the model's capabilities and constraints, especially regarding performance in high-dimensional settings.

W4: For high-dimensional NN search, the paper compares its model only against the data-independent LSH baseline. However, recent advancements in unsupervised learning-to-hash methods provide competitive alternatives that also leverage data distribution knowledge, for example [1] and [2]. Including comparisons with these modern hashing approaches could help position the framework’s performance more accurately within the field and reveal any advantages or limitations when competing with other data-aware methods.

W5: The transition from NN search to frequency estimation is somewhat abrupt and lacks a unified explanation of how these tasks fit within the general framework. The two tasks use different backbones and have different outputs for each component. While both applications involve end-to-end data structure learning, the framework’s flexibility for adapting to different problems is not fully articulated. Clarifying the underlying principles that guide network design for new tasks within the framework would enhance its usability.

References:

[1] Lin, Kevin, et al. "Learning compact binary descriptors with unsupervised deep neural networks." Proceedings of the IEEE conference on computer vision and pattern recognition. 2016.

[2] Jin, Sheng, et al. "Unsupervised semantic deep hashing." Neurocomputing 351 (2019): 19-25.

**Questions:**

1. Could the author discuss the justification for why a transformer with $O(n^2)$ complexity is necessary, and why using such a network is still feasible and beneficial for the NN search task? (W1)

2. Could the author demonstrate how well the model performs on real-world data distributions in addition to MNIST by including other real-world datasets that may not follow synthetic patterns? (W2)

3. Could the author clarify the implementation details and/or provide the code for the experiments? (W3)

4. Is there a reason that more recent learning-to-hash methods were not included as baselines for the high-dimensional nearest neighbor search experiment? (W4)

5. The application of the framework to nearest neighbor search and frequency estimation seems somewhat disconnected. Could the authors elaborate on the principles that unify these applications within the framework? Could the authors provide a set of design principles or guidelines for adapting the framework to other tasks? This would make it clearer how researchers can extend this work. (W5)

---

> ### Author Response · Authors · 2024-11-19
> **Response [1/2]**
>
> We thank the reviewer for your detailed feedback and appreciate the time and effort you’ve dedicated to evaluating our work!
>
> We’ve addressed your specific questions below, however, we also kindly ask you to read our general response to all reviewers where we clarify the motivation and contributions of our work, discussing how it can provide useful insights for data structure design and avenues for scaling.
>
> >Considering the NN search problem has a trivial solution with O(n)complexity, this constraint raises concerns about the framework's practicality in real-world applications where much larger datasets are typical.
>
> ****
>
> We’d like to clarify what may be a misunderstanding. For a given dataset, the data-processing network only needs to be executed once with an O(n^2) cost (this could be made even cheaper as we discuss in our next response), i.e. there is a fixed construction cost. Thereafter, the query models are restricted to M lookups where M < N and so each query to the data structure costs O(M).
>
> >Could the author discuss the justification for why a transformer with O(n2) complexity is necessary, and why using such a network is still feasible and beneficial for the NN search task? (W1)
>
> We chose to use a transformer with O(n^2) complexity in order to keep the model relatively general. As you’ve noted, this may not always be necessary. For instance, high-dimensional hashing-based methods could be learned with a less complex model. However, data structures like k-d trees have a higher construction complexity O(dnlogn). Our framework does not require using any specific model so a less (or more) complex model can trivially be substituted in for the transformer. As we mention in our general response on scaling, one way to enable scaling to larger instances would be to lower the complexity of the data-processing model (e.g. using a linear transformer).
>
> >Could the author demonstrate how well the model performs on real-world data distributions in addition to MNIST by including other real-world datasets that may not follow synthetic patterns? (W2)
>
> Yes, we include additional experiments run on the SIFT (128 dim) and Fashion-MNIST (100 dim) datasets and compare them to learning-to-hash methods. Our model performs competitively with these learning-to-hash baselines, only slightly underperforming them. We didn’t do any hyperparameter tuning in order to respond quickly to reviewers but we’ll obtain final results (with tuning) for the camera-ready version. We also show an example of how the data-processing model learns to transform the Fashion-MNIST data by organizing the images by class.
>
> In addition to the HighD NN experiments, we also include an additional frequency estimation experiment where we show how insights we learned from our model suggested improvements to the CountMinSketch algorithm which lead to improvements on a large-scale real world IP dataset.
>
> Please see Appendix G of the updated revision for more details on both the NN and frequency estimation experiments.
>
> >Could the author clarify the implementation details and/or provide the code for the experiments? (W3)
>
> >- (1) The paper uses NanoGPT, a decoder-only transformer that typically requires discrete tokens for input and output. However, the paper doesn’t clearly explain how continuous data inputs are tokenized or how outputs are transformed into the scalar ranking oi values. More detail on the embedding and decoding process for continuous data is necessary to understand the data flow and model structure.
>
>   - We apologize for the confusion. There is no specific reason as to why we chose NanoGPT other than that there is an open source implementation available that uses flash attention which allows for faster training.  Any other transformer model should work. Instead of using a discrete tokenizer, we have input and output projection layers (Linear layer) that project the input to the hidden dimension (64) and from the hidden dimension to the scalar values.
>
> >- (2) In experiments involving CNNs, essential details about the CNN architecture, such as the number of layers, kernel sizes, or feature extraction strategies, are missing. Including these specifications would make it easier to interpret the model's capabilities and constraints, especially regarding performance in high-dimensional setting**
>
>   - We use a simple CNN model with the following architecture:
>
> <!---->
>
>     Conv2d(in_channels=1, out_channels=32, kernel_size=3, stride=1, padding=1)
>     ReLU()
>     MaxPool2d(kernel_size=2, stride=2, padding=0)
>     Conv2d(in_channels=32, out_channels=64, kernel_size=3, stride=1, padding=1)
>     ReLU()
>     MaxPool2d(kernel_size=2, stride=2, padding=0)
>     Linear(in_features=64*7*21, out_features=128)
>     ReLU()
>     Linear(in_features=128, out_features=1)
>
> We plan to fully open source the code before publication.

---

> ### Author Response · Authors · 2024-11-19
> **Response [2/2]**
>
> >Is there a reason that more recent learning-to-hash methods were not included as baselines for the high-dimensional nearest neighbor search experiment? (W4)
>
> We excluded learning-to-hash methods as regular LSH is already a strong baseline in the 30D setup we considered and hashing-based methods are not designed to handle our MNIST setting. Specifically, in our MNIST setup the metric is not given but implicitly provided via nearest neighbor supervision. This means our model needs to learn the metric space, which learning-to-hash methods are not designed for. However, in response to your question and to highlight our models strengths in this regime, we compare with an optimal hash function - splitting the 200 number universe evenly into K clusters where the first N/K elements are assigned to cluster 1, the second N/K  elements to cluster 2 and so forth. To execute a NN search, a query is mapped to its corresponding bucket and the element closest to this number (defined over this 1D label space) is chosen as the NN. When N=50 and M=7 we find that the optimal K is 8. To be clear, this is an unfair comparison as our model has to learn features that capture the underlying metric space but in this case it's provided directly. However, this optimal hashing setting should still underperform our model in this regime as it does not use an adaptive query algorithm which is necessary for achieving the O(log N) performance our model recovers. See Figure 10 in our updated revision for the comparison between our model and this hashing baseline (LSH).
>
> This raises an important point we’d like to emphasize about our model - by directly learning features, a data structure and a query algorithm end-to-end it can better exploit the low-dimensional structure of this problem. Specifically, learning-to-hash methods simply cluster the points by their hash code and then search every element within the corresponding hash bucket. Our method, however, can learn an adaptive query algorithm (in this case something like 1D interpolation search over the learned features), thereby identifying the NN with fewer lookups than a hashing-based method.
>
> We do, however, include comparisons to learning-to-hash methods for the additional high-dimensional SIFT/FashionMNIST experiments we ran as these are fairer comparisons (See Appendix G.2 of the paper).
>
> >The application of the framework to nearest neighbor search and frequency estimation seems somewhat disconnected. Could the authors elaborate on the principles that unify these applications within the framework? Could the authors provide a set of design principles or guidelines for adapting the framework to other tasks? This would make it clearer how researchers can extend this work. (W5)
>
> Among other reasons, we chose to apply our framework to both of these settings to highlight the versatility of the framework and demonstrate how different data structure problems can be modeled by the same framework. In both settings, raw data first needs to be added to a data structure and then the data structure needs to be efficiently accessed to respond to some query. In our nearest neighbor setup, all the raw data is ingested together at one time (batch setting). However, in the frequency estimation setting, a single raw data point arrives at a time (streaming setting). Our framework is general and can handle both the batch and streaming setting, though different model architectures may be more appropriate for a given setting. For instance, in the streaming setting we use an MLP as the data-processing network as there is only a single point to process.  Both problems then require efficiently querying the data structure which we enforce by restricting the number of lookups the query models can make into the data structure.
>
> The framework is general in that it is designed to handle problems that require efficiently responding to queries about some collection of data. The queries access a _data structure_ and efficiency is controlled by trading off the query complexity (number of lookups into the data structure) and the space complexity (size of the data structure). If other researchers had a candidate problem that could be framed like so (In section 3.2  we provide several examples), we believe our framework could be applicable. Specifically, the problem can be modeled with both a data-processing network and query network and could use similar principles we develop to control the efficiency of the data structure (e.g. tokenizing and restricting its size to control space complexity and using similar sparsity techniques on lookup vectors to enforce query complexity). The choice of architecture for each network would depend on the specific setting. For instance, for graph data-structures it may be more efficient to use a graph neural network for the data-processing network.&#x20;
>
> We hope this response answers your questions and addresses your concerns. Please let us know if you have any other questions!

---

> > ### Comment · Reviewer_mekY · 2024-11-24
> >
> > Thank you for your detailed response! It addressed some of my concerns, and I have some follow-up questions and request for clarification:
> >
> > **Regarding (W1):**
> >
> > The indexing cost of $O(n^2)$ remains prohibitively high, especially when considering the additional overhead introduced by the model parameters. This raises questions about the practicality of adopting such a costly approach when its effectiveness appears only comparable to common baselines.
> >
> > While the authors suggest that alternative models, such as linear transformers, could reduce costs, they do not present new results to substantiate these claims. The lack of empirical evidence for such improvements weakens the overall argument and leaves the framework's feasibility for large-scale, real-world applications unconvincing.
> >
> > **Regarding (W2):**
> >
> > Thank you for including additional results on the SIFT (128-dim) and Fashion-MNIST (100-dim) datasets and for comparing them with learning-to-hash methods. However, in Fig. 26, the results for the proposed method are not particularly strong, ranking only 3rd. The explanation that hyperparameter tuning was omitted to respond quickly to reviewers is difficult to accept, as it leaves uncertainties about the model's true potential. It would be more convincing to provide tuned results, even in subsequent revisions.
> >
> > **Response (W5):**
> >
> > The response primarily reiterates points from the original paper without offering substantial new insights or clear illustrations of how the framework unifies the two problems. The explanation still gives the impression of two separate methods rather than a truly cohesive framework.
> >
> > Additionally, as highlighted by Reviewer 5rfb, generalizing this framework to other data structure problems is not straightforward. It requires considerable trial-and-error, along with expertise in the classical methods for the specific data structures being addressed. This lack of explicit design principles or guidelines limits the framework’s scalability, making it challenging for researchers to adapt it to new problems without substantial domain knowledge and iterative experimentation.

---

> > > ### Author Response · Authors · 2024-11-25
> > >
> > > Thank you for following-up! We’ve addressed each of these points below:
> > >
> > > **Regarding (W1):**
> > >
> > > We’ve included additional experiments as evidence that cheaper alternatives to a (quadratic-attention) transformer can be substituted in our framework. Specifically, we ran experiments in both 2D and High-D (FashionMNIST) where we used a linear transformer (Performer Model [1]) instead of the vanilla Transformer. We use the same model hyper parameters that we used for the Transformer. We chose the 2D setting as in this case adaptivity is beneficial for the query-algorithm and so learning a data structure that can support adaptive queries could require a higher indexing cost. In the high D setting, adaptivity is less crucial (as supported by the strong performance of the learning-to-hash baselines), however, we wanted to see if a less complex model could still support high-dimensional data. We found that in 2D, the linear transformer still outperformed the KD tree baseline significantly and only slightly underperformed the vanilla transformer. This could be, as we mentioned, due to the fact that modelling more interactions (with N^2 attention) can discover a (slightly) better data structure for adaptive querying, though more investigation would be needed to confirm this. In High D, our model performs as well as the vanilla transformer. We believe these results support our claim that cheaper alternatives to quadratic attention can be used in certain settings.
> > >
> > > Please find the results in App. G.3 Figure 28
> > >
> > > **Regarding (W2):**
> > >
> > > We ran additional learning-rate hyper parameter tuning on SIFT (we focused our resources on SIFT as we underperformed more on this dataset) and were able to recover comparable performance to the other learning-to-hash baselines (we’ve updated the plots in the new revision). We will additionally run hyper-parameter tuning for FashionMNIST after the rebuttal period. To be clear, we do not expect our model to outperform these baselines in this setting as it is unclear that query adaptivity should be beneficial here. Rather, we include these results to further emphasize that our E2E model can recover reasonable solutions in a variety of settings - even when compared to carefully hand-designed solutions. In our view, the MNIST result is more interesting as it demonstrates our model’s ability to learn low-dimensional features that enable more efficient, adaptive querying, something that prior methods cannot do.
> > >
> > > **Regarding (W5):**
> > >
> > > We acknowledge that additional design decisions beyond our framework need to be made to support new types of problems, as is the often the case in ML applications where some domain knowledge is also required. Our main point, however, is that data structure problems do share common structure and constraints, and that our framework offers one possible way of approaching these types of problems. As we mentioned, the commonalities include having both a data-processing/query algorithm and the need to enforce space/query-time complexity constraints on the algorithms. Crucially, enforcing query-time complexity constraints requires learning sparse algorithms and we propose one promising way to do this (training with noise). More broadly, our hope with the second study into frequency estimation, and the other examples we discussed, was to encourage more research into end-to-end design of data structures.
> > >
> > > We hope this addresses your concerns and please let us know if you have any other questions!
> > >
> > > [1] Krzysztof Choromanski, Valerii Likhosherstov, David Dohan, Xingyou Song, Andreea Gane, Tamas
> > > Sarlos, Peter Hawkins, Jared Davis, Afroz Mohiuddin, Lukasz Kaiser, David Belanger, Lucy
> > > Colwell, and Adrian Weller. Rethinking attention with performers, 2020.

---

> > > > ### Author Response · Authors · 2024-11-26
> > > >
> > > > We’ve finished tuning our model on FashionMNIST as well and have updated the revision with the new results.

---

> > > > > ### Comment · Reviewer_mekY · 2024-11-28
> > > > >
> > > > > Thank you for the detailed new experiments and updates. The results with the linear transformer (W1) indeed strengthen the claim and provide valuable insights, and I appreciate the additional hyperparameter tuning for SIFT (W2), which improves the accuracy. These improvements address key concerns, and I will raise my score from 3 to 5.
> > > > >
> > > > > However, I remain cautious about the generalizability of the framework (W5), as the dependency on domain-specific design choices limits its broader applicability. Additionally, the lack of discussions of efficiency and scalability issues leaves critical questions unanswered for the applications requiring large-scale data. These concerns prevent me from assigning a higher score at this time.

---

> ### Author Response · Authors · 2024-11-29
>
> Thank you for your response! We address your remaining concerns below.
>
> >However, I remain cautious about the generalizability of the framework (W5), as the dependency on domain-specific design choices limits its broader applicability.
>
> While we have already discussed how higher-level elements of our framework are applicable to new problems beyond those we explore, we’d like to clarify what would be required in terms of model architectures as we believe this may have been unclear from our prior responses.
>
> For new data structure problems, such as those we mention in section 3.2, one could certainly use the same architectures we have used, i.e. for the data-processing model, using a transformer for batch-setting/MLP for streaming setting and using MLPs for the query-model. These architectures are relatively general and therefore require minimal adaptation. To be more specific,  we discuss how each of the problems we propose in Section 3.2 can be tackled with our framework and minimal problem-specific knowledge (we focus on the batch setting as this is the more common one though similar insights apply to the streaming setting):
>
> **Graph data structures:** A permutation-equivariant architecture such as a transformer (with no positional encodings) or a graph-neural network should be used to represent the graph in the data-processing network. Note that it has already become commonplace to use transformers to represent graphs [5, 6]. The output size of the model would be constrained to control the space complexity of the learned data structure. This is accomplished by either ignoring certain output tokens when less space then the input size is desired or adding additional tokens when more space is required, as we do in our extra space experiments. The same MLP query architecture we proposed should suffice.
>
> **Sparse matrices:** As matrices are simply grids of numbers, transformer would be appropriate as well with each token representing either a row/column of the matrix or a single element. The output size of the model would be constrained to control the space-complexity of the learned data structure. The same MLP query architecture we proposed should suffice.
>
> **Learning statistical models**: In this case, the data-processing network which operates over datasets should likely be permutation-invariant as there is no canonical ordering to an IID dataset so a transformer would suffice. Again, the output size of the model would be constrained to control the size of the learned model (e.g. decision tree). The MLP query architecture would be an appropriate choice for the query model.
>
> In summary, we believe minimal problem-specific knowledge is required to adapt our framework to new types of data structure problems. While specialized architectures do exist for different domains (e.g. graph neural networks), transformers can certainly be used as a starting point and are currently applied to many different types of inputs [3, 4, 5]. There is also work [1, 2] that is trying to build general architectures for structured inputs/outputs. Advances in this direction could lead to more general architectures for data structures as well.
>
> We will definitely include this more detailed discussion in our next revision.
>
> >Additionally, the lack of discussions of efficiency and scalability issues leaves critical questions unanswered for the applications requiring large-scale data.
>
> In our original submission we included a section on scaling and limitations (Appendix F), however, we could certainly update this section to reflect our discussion with a more detailed analysis addressing avenues to improve efficiency as well (e.g. Linear Transformers). We will also move this section to the main paper to increase transparency. We will make these changes after the rebuttal period as currently we cannot upload a new revision.
>
> We hope this addresses your concerns. Please let us know if you have any other questions.
>
> [1] Jaegle, A., Gimeno, F., Brock, A., Vinyals, O., Zisserman, A., & Carreira, J. (2021, July). Perceiver: General perception with iterative attention. In International conference on machine learning (pp. 4651-4664). PMLR.
>
> [2] Borgeaud, Jean-Baptiste Alayrac, Carl Doersch, Catalin Ionescu, David Ding, Skanda Koppula et al. "Perceiver io: A general architecture for structured inputs & outputs." arXiv preprint arXiv:2107.14795 (2021).
>
> [3] Zhao, H., et al (2021). Point transformer. In Proceedings of the IEEE/CVF international conference on computer vision (pp. 16259-16268).
>
> [4] Dosovitskiy, A. (2020). An image is worth 16x16 words: Transformers for image recognition at scale. arXiv preprint arXiv:2010.11929.
>
> [5] Dwivedi, V. P., & Bresson, X. (2020). A generalization of transformer networks to graphs. arXiv preprint arXiv:2012.09699.
>
> [6] Ying, C. et al (2021). Do transformers really perform badly for graph representation?. Advances in neural information processing systems, 34, 28877-28888.

---

> > ### Author Response · Authors · 2024-12-03
> >
> > Hi, thanks again for taking the time to review our submission!
> >
> > As the discussion period comes to an end today, we wanted to check in to make sure we’ve answered all of your questions appropriately. We are happy to try to address any other comments in the time remaining.

---

### Official Review · Reviewer_5rfb · 2024-10-31

**Soundness:** 4
**Presentation:** 4
**Contribution:** 3
**Rating:** 8
**Confidence:** 3

**Summary:**

This paper proposes using deep learning to create data structures. The model learns to rank, i.e. learns a mapping of data element to scalar. The model optionally also can output further auxiliary data to help accelerate the data structure. Querying is performed by M lookups into the data structure, where each lookup is chosen by a separate MLP. The Mth lookup is the query result.

This paper looks at the following data structure problems:
* 1D nearest neighbors
* 2D, 30D, and MNIST (high-dimensional with structure) nearest neighbors
* Frequency estimation, done with a separate model architecture.

The paper shows that the models are effective and adapt to the data distributions presented to it during training.

**Strengths:**

* Impressive visualizations and demonstrations of distribution awareness: the plots showing k-d tree resemblance and effectiveness over binary search on a 1D Zipfian distribution were especially impressive.
* Novel idea with good execution
* Paper is well organized and written, and easy to follow

**Weaknesses:**

* Impractical for now: only works for small datasets (100 and 500 elements were tested).
* Still requires a moderate amount of inductive bias: for example, the frequency estimation architecture is very different than the nearest neighbors one, and does not seem obvious to me. It seems that generalizing this approach to new data structure problems isn't entirely trivial and requires at least some trial-and-error, as well as expertise in how classical approaches to the data structure may work.
* In short, the main advantage of this paper's idea is its ability to adapt to data distribution, but the real-world benefit of this is hampered by the fact that for now, this work cannot scale to datasets large enough for distribution-fitting to really show impact (especially when considering the performance impact of MLPs vs, for instance, comparing integers in a BBST).
* No code release, especially disappointing given that the training tasks require only synthetic data and MNIST, and involve small models.

**Questions:**

Do you have code to share to replicate your results?

---

> ### Author Response · Authors · 2024-11-19
>
> We thank the reviewer for your feedback and appreciate the time and effort you’ve dedicated to evaluating our work!
>
> We kindly ask the reviewer to read the general response we posted as they address your comments regarding the scalability of learning data structures end-to-end.
>
> >Still requires a moderate amount of inductive bias: for example, the frequency estimation architecture is very different than the nearest neighbors one, and does not seem obvious to me. It seems that generalizing this approach to new data structure problems isn't entirely trivial and requires at least some trial-and-error, as well as expertise in how classical approaches to the data structure may work.
>
> We acknowledge that the data-processing architectures are different for the NN and frequency estimation problems (the query architectures remain the same), and that other problems may benefit from problem-specific architectures however we still believe that the general framework we’re proposing can provide a scaffold for new data structure problems. Specifically, the problem can be modeled with both a data-processing network and query network and could use similar principles to control the efficiency of the data structure (e.g. tokenizing and restricting its size to control space complexity and using similar sparsity techniques on lookup vectors to enforce query complexity).
>
> >Do you have code to share to replicate your results?
>
> We will definitely fully open-source the code before publication.
>
> We hope this response answers your questions, please let us know if you have any other questions!

---

> > ### Comment · Reviewer_5rfb · 2024-11-24
> >
> > Thank you for your response. I maintain my score, and generally disagree with the critiques presented by other reviewers. I see the fact that end-to-end deep learned data structures work at all as an interesting result, and I think other reviewers' expectations of these learned structures outperforming existing, classically designed, SOTA as an excessively high bar. Furthermore, I see potential for this work to help with classical data structure design by "distilling" insights from E2E learned structures into inductive bias that can be baked into standard data structures.

---

### Official Review · Reviewer_NYBv · 2024-11-01

**Soundness:** 3
**Presentation:** 4
**Contribution:** 1
**Rating:** 3
**Confidence:** 5

**Summary:**

The article proposes an end-to-end learning framework for data structures, and implement it in the cases of sorting, nearest neighbor search, and frequency estimation. The method is based on joint learning of a data processing network (8 layer transformer model in the implementation of the article) and query processing networks (MLPs in the implementation of the article). The data processing network is used to learn an index structure by ranking and then sorting the original data set. The query processing networks are trained to predict the correct output (e.g., the nearest neighbor of the query point) given the output of the data processing network and the query execution history (i.e., the outputs of the earlier lookups).

The proposed framework differs from the earlier work on learning-augmented data structures since it does not require specifying the data structure that is used, but is completely end-to-end. The authors test they approach on very small ($N \approx 100$, $d =1,2,30$) simulated data sets, and show that it can match the performance of or outperform simple worst-case algorithms and data structures, such as binary search, k-d trees, and locality-sensitive hashing (LSH).

**Strengths:**

The proposed framework is novel as far as I know. It is curious that the models learn to replicate binary search and a k-d tree structure. The article is well-written and easy to follow.

**Weaknesses:**

The article is very exploratory. The data sets are simulated and several magnitudes smaller ($N \approx 100$) and have smaller dimensionality ($d=1,2,30$) than the ones used in the practical applications of nearest neighbor search, and the baseline algorithms are very elementary. While the results have some curiosity value, it seems very unlikely that the proposed method could be (a) scaled to the data sets that are used in the real applications of nearest neighbor search (typically, $N>1000 000$ and $d \in [100,1000]$ ) AND (b) match the performance of SOTA approximate nearest neighbor search algorithms, such as ScaNN (Guo et al., 2020) and HNSW (Malkov & Yashunin, 2018). And even if it were be possible, this would require enormous computational resources.

Approximation algorithms, such as approximate nearest neighbor search, are used to _save_ computational resources and speed up ML pipelines that use computationally heavy components. For instance, ANN search has recently been used for approximate attention computation in transformer models (Kitaev et al. 2020; Roy et al., 2021), and to speed up inference in retrieval-augmented generation (RAG) (Borgeaud et al., 2022; Lewis et al., 2020).

Thus, I really fail to see how adding a computationally intensive component, such as a transformer model, to the approximation pipeline could be a useful or valuable contribution, even though it resulted in a slightly decreased query latency compared to the SOTA ANN algorithms (which the article is not even close to demonstrating). In contrast, the SOTA ANN libraries, such as FAISS and ScaNN (Douze et al., 2024) use elementary tools, such as $k$-means clustering, for indexing. This is because they have to scale to billion-scale data sets, and thus the index construction time has to be reasonable.

Borgeaud, Sebastian, et al. "Improving language models by retrieving from trillions of tokens." International conference on machine learning. PMLR, 2022.

Douze, Matthijs, et al. "The faiss library." arXiv preprint arXiv:2401.08281 (2024).

Guo, Ruiqi, et al. "Accelerating large-scale inference with anisotropic vector quantization." International Conference on Machine Learning. PMLR, 2020.

Kitaev, Nikita, Lukasz Kaiser, and Anselm Levskaya. "Reformer: The Efficient Transformer." International Conference on Learning Representations. 2020.

Lewis, Patrick, et al. "Retrieval-augmented generation for knowledge-intensive nlp tasks." Advances in Neural Information Processing Systems 33 (2020): 9459-9474.

Malkov, Yu A., and Dmitry A. Yashunin. "Efficient and robust approximate nearest neighbor search using hierarchical navigable small world graphs." IEEE transactions on pattern analysis and machine intelligence 42.4 (2018): 824-836.

Roy, Aurko, et al. "Efficient content-based sparse attention with routing transformers." Transactions of the Association for Computational Linguistics 9 (2021): 53-68.

**Questions:**

I do not have any particular questions for the authors.

---

> ### Author Response · Authors · 2024-11-19
>
> We thank the reviewer for your feedback and appreciate the time and effort you’ve dedicated to evaluating our work!
>
> We kindly ask the reviewer to review the general response we posted as we believe there may be a misunderstanding around the motivation and contribution of our work. Our intentions with this work were neither to scale to very large datasets nor to beat existing SOTA approaches in these regimes (though we discuss how to further scale up the model in our response), but rather to understand what neural networks trying to learn data structures end-to-end are capable of doing, as a step towards using them for gaining insight into data structure design.
>
> We hope this response addresses your concerns. Please let us know if you have any other questions!

---

> > ### Comment · Reviewer_NYBv · 2024-11-20
> >
> > Thank you for your response and for clarifying your contribution in the general response.
> >
> > I want to clarify that my main issue with the article is not that it is exploratory (for instance, all research in quantum computing is also exploratory for the time being), but that the research direction it explores does not seem valuable to me. As an expert on design of data structures, I do not see the utility of the article for that field. To summarize my review, approximation algorithms and data structures are currently used to speed up deep learning applications, and adding computationally expensive components to the approximation pipeline seems counterproductive. However, I agree with the author's response that there is a minor contribution in demonstrating the learning capabilities of neural networks, but in my opinion this contribution does not yet pass the acceptance threshold.

---

> ### Author Response · Authors · 2024-11-21
>
> Thank you for your response. As a reminder, we are not claiming that our specific method should be used as a direct substitute in existing data structure pipelines. Rather, we show that our framework can be used to gain insight into data structure design, and that these insights can be scaled and applied in realistic settings (e.g. our augmented CountMinSketch algorithm in Appendix G.1).
>
> We’d like to re-emphasize that this is the first work that explores E2E learning of data structures (to the best of our knowledge) and so it seems premature to claim that they cannot be useful for advancing our understanding of data structures. In fact, augmenting data structures with predictions from neural networks has already been shown to be effective [1, 2, 3]. We’ve also discussed multiple avenues for future work to reduce the computational cost of learning data structures E2E.
>
> [1] Tim Kraska, Alex Beutel, Ed H Chi, Jeffrey Dean, and Neoklis Polyzotis. The case for learned
> index structures. In Proceedings of the 2018 international conference on management of data,
> pp. 489–504, 2018.
>
> [2] Honghao Lin, Tian Luo, and David Woodruff. Learning augmented binary search trees. In International Conference on Machine Learning, pp. 13431–13440. PMLR, 2022a.
>
> [3] Yihe Dong, Piotr Indyk, Ilya Razenshteyn, and Tal Wagner. Learning space partitions for nearest
> neighbor search. arXiv preprint arXiv:1901.08544, 2019.

---

### Official Review · Reviewer_ApFo · 2024-11-04

**Soundness:** 2
**Presentation:** 3
**Contribution:** 2
**Rating:** 5
**Confidence:** 3

**Summary:**

This paper aims at discovering data structures for nearest neighbor search in an end-to-end manner. The method proposed in this paper employ two networks, i.e., data-processing network and query execution network, for the purpose. The proposed method is pretty costly such that only very few data points are included in the experiments, which makes the method difficult to be evaluated for its practical usefulness.

**Strengths:**

S1. This paper studies an interesting problem.

S2. The method may work.

S3. It is a novel work.

**Weaknesses:**

W1. The method proposed is too costly for finding such data structures.

W2. The experiments are not sufficient to support its practical usefulness.

W3. The data structures found seems too simple, i.e., sorting and K-d trees.

**Questions:**

Q1. Is it possible to verify the effectiveness of the method with a large data that approaches the real applications?

---

> ### Author Response · Authors · 2024-11-19
>
> We thank the reviewer for your feedback and appreciate the time and effort you’ve dedicated to evaluating our work!
>
> >Q1. Is it possible to verify the effectiveness of the method with a large data that approaches the real applications?
>
> We kindly ask the reviewer to read the general response we posted as they address your comments regarding the scalability and cost of learning data structures as well as the practical usefulness of our results. We also refer the reviewer to Appendix G of our updated revision where we include additional experiments showing how our model can provide useful, scalable insights and be applied to more realistic data.
>
> >W3. The data structures found seem too simple, i.e., sorting and K-d trees.
>
> Sorted lists, k-d trees, and LSH data structures are among the most widely used and fundamental data structures and so it is significant that our model can discover useful data structures. Moreover, beyond 1-dimension, data structure design is particularly  challenging as there is no clear notion of ordering so recovering any useful data structure is impressive.
>
> We also emphasize that it’s not trivial that any useful data structure can be recovered given that both the query and data structure model are being trained together from scratch - one can imagine that there exist many local minima where both models cannot make progress. For instance, in 1D, it is quite surprising that the query model can learn an algorithm such as binary search while also learning the correct data structure (a sorted list) to operate over. Moreover, both models are searching over a discrete and combinatorial space which poses even further optimization challenges.
>
> We hope this response answers your questions and addresses your concerns. Please let us know if you have any other questions!

---

> > ### Author Response · Authors · 2024-11-25
> >
> > Hi, thanks again for taking the time to review our submission!
> >
> > As the discussion period comes to an end tomorrow, we wanted to check in to make sure we’ve answered all of your questions appropriately. In addition to our general response where we clarify our motivation and contributions, we invite you to read Appendix G of our latest revision that includes new experiments with more real-world data and computationally-cheaper alternatives for our framework.
> >
> > We are happy to try to address any other comments in the time remaining.

---

### Author Response · Authors · 2024-11-19
**General Response To All Reviewers [1/3]**

Thank you to all reviewers for your thoughtful feedback and valuable suggestions! We appreciate the time and effort you’ve dedicated to evaluating our work. We've put together a general response to all reviewers and have also responded to individual questions directly.

## Our Motivation
There appears to be a misunderstanding regarding the motivation of our work. As reviewers have noted, one possible motivation for learning data structures end-to-end is to replace hand-designed database systems with learned data structures at scale. However, an equally important motivation is to gain insights into data structure design—both to develop better data structures and to understand the broader landscape of possibilities. While our work serves as a starting point for both perspectives, our primary focus was the latter. Our intention was neither to scale to very large datasets nor to beat existing standard approaches. Instead, we aimed at understanding what neural networks trying to learn data structures end-to-end are capable of doing, as a step towards using them for gaining insight into data structure design. To clarify the significance and novelty of our work, we first restate our main contributions and their relevance. Following this, we will address the issue of scaling end-to-end learning. We will clarify these points in the final version of the paper.

## Primary contributions

### 1. **We show that neural networks can be trained to discover data structures end-to-end.**

Our work is (to the best of our knowledge) the first work that explores end-to-end learning of data structures. While end-to-end learning has become the dominant paradigm in deep learning, it’s not obvious that this should work for data structures, nor is it clear how to structure this problem. Discovering a data structure requires both finding an algorithm to construct the data structure _and_ an algorithm to query the data structure. In contrast to prior work that either learns algorithms using explicit supervision\[5] or mainly focuses on learning a single algorithm at a time\[1, 3, 5], we show that it’s possible to learn two algorithms in tandem without explicit supervision. It might be hard to imagine this joint learning process even getting off the ground. For instance, if the data-processing network produces a random garbled function of the dataset, we cannot hope the query model to do anything meaningful. This is further amplified by the combinatorial nature of how the query model accesses the data structure. Therefore, the fact that well-known and widely used data structures (e.g. sorted-lists, KD trees, locality sensitive hashing) can be recovered, we believe, is surprising in its own right.

---

> ### Author Response · Authors · 2024-11-19
> **General Response To All Reviewers [2/3]**
>
> ### 2. **We show learning data structures end-to-end can provide useful insights into data structure design**
>
> As discussed above, the learned data structures can serve as a source of insight for data structure design. There are several points in this vein that are worth highlighting:
>
> **The learned data structures and insights do scale:** While we don’t demonstrate the scalability of the model itself, the learned data structures and algorithms do in fact scale (e.g. LSH, KD Trees, Binary Search, etc). This suggests that one way one could use our framework is by first learning a data structure, followed by reverse-engineering it and applying it to larger settings.
>
> Even when it is not possible to reverse-engineer the complete data structure, one can still discover insights to improve existing data structures. For instance, in the frequency estimation problem, we observed that the model performs well by using update deltas smaller than 1. We used this insight to create a modified version of CountMinSketch with smaller update deltas and applied it to a much larger and real world dataset and show that it outperforms traditional CountMinSketch by a factor of two in some regimes. (See Appendix G.1 of the updated revision for more details)
>
> **Distributional awareness:** We show that distribution-dependent data structures and query algorithms emerge even at small scale. This suggests that one could use our framework to determine a lower-bound on performance for a distribution-dependent data structure.
>
> **Understanding the landscape of data structures:** There are many fundamental open questions around understanding the landscape of possible data structures. For example, in low-dimensional nearest neighbor search, methods like KD-trees use O(n) space and guarantee O(log n) query complexity, where n is the number of input points. However, their query time scales exponentially with dimension. Conversely, methods like locality-sensitive hashing (LSH) for approximate nearest neighbor search in high dimensions use a reasonable amount of space and guarantee $O(n^{(1 - \delta)})$ query complexity, where \delta reflects the correlation between the query point and its nearest neighbor. While LSH avoids the exponential blow-up with dimension, the presence of $\delta$ in the exponent means performance degrades quickly as the correlation diminishes, unlike KD-trees. Are there any data structures in between that can achieve the best of both worlds? Our framework for end-to-end learning of data structures can possibly be used as a tool to make progress on such fundamental questions.
>
> At the very least, one might hope that when high-dimensional data lies on low-dimensional manifolds (a common scenario), it would be possible to learn the low-dimensional structure and apply KD-tree-like methods to achieve logarithmic query complexity. We demonstrate that this is possible using our framework in the MNIST experiment, where our model identifies the relevant low-dimensional 1d feature in MNIST images (i.e. the relative ordering of numbers), sorts the data according to this feature, and applies binary search to achieve logarithmic query complexity.
>
> ### 3. **Our proposed framework can be useful for modeling other data structure problems beyond NN/frequency estimation.**
>
> We believe that the general framework we’re proposing can provide a useful scaffold for other data structure problems beyond those that we examine in our work (in section 3.2 we provide several examples). Specifically, these types of problems can be modeled with both a data-processing network and query network and could use similar principles to control the efficiency of the data structure (e.g. tokenizing and restricting its size to control space complexity and using similar sparsity techniques on lookup vectors to enforce query complexity).

---

> > ### Author Response · Authors · 2024-11-19
> > **General Response To All Reviewers [3/3]**
> >
> > ## **Can end-to-end learning be scaled to larger datasets?**
> >
> > While we believe the models we use (transformer + MLP models) can support much larger datasets simply by scaling parameter counts and training time, we are not suggesting this is the most efficient way to do so, rather that it can be done. As reviewers have pointed out, the quadratic-attention operation of the transformer is not always necessary to learn useful data structures and can likely be substituted for a cheaper alternative such as linear attention or a single MLP depending on the complexity of interactions between data points one wishes to be able to model. We chose to use quadratic-attention to leave the model relatively general at first. Additionally, recent advancements have demonstrated the feasibility of scaling large language models (LLMs) to handle context sizes exceeding 1 million tokens\[6], underscoring the potential for attention-based models to operate at much larger scales. In general, we believe that numerous other inductive biases can be added within our proposed framework to support E2E learning at scale.
> >
> > We also point to other works focused on algorithm learning and discovery that started at small scale with more synthetic setups \[1, 2, 3] and inspired follow-up work that then focused on scaling up the initial set of ideas. We hope our work can serve a similar purpose - demonstrating the value in learning data structures E2E - and inspire future work to develop methods for better scaling.
> >
> > \[1] [NEURAL COMBINATORIAL OPTIMIZATION WITH REINFORCEMENT LEARNING](https://arxiv.org/pdf/1611.09940)
> >
> > \[2] [NEURAL ARCHITECTURE SEARCH WITH REINFORCEMENT LEARNING](https://openreview.net/pdf?id=r1Ue8Hcxg)
> >
> > \[3] [Learning a SAT Solver from Single-Bit Supervision](https://openreview.net/forum?id=HJMC_iA5tm)
> >
> > \[4] [Learning to learn by gradient descent by gradient descent](https://proceedings.neurips.cc/paper_files/paper/2016/file/fb87582825f9d28a8d42c5e5e5e8b23d-Paper.pdf#page=7.36)
> >
> > \[5] [\[1910.10593\] Neural Execution of Graph Algorithms](https://arxiv.org/abs/1910.10593)
> >
> > \[6] [Gemini: https://blog.google/technology/ai/google-gemini-next-generation-model-february-2024/#sundar-note](https://blog.google/technology/ai/google-gemini-next-generation-model-february-2024/#sundar-note)

---

### Author Response · Authors · 2024-12-03
**Summary of discussion period**

Hi,

We thank the reviewers for your feedback! We wanted to provide a short summary of some of the concerns that were raised during the discussion period and how we addressed them.

**Experiments with Realistic Data**

We added additional experiments with high-dimensional datasets for the nearest neighbour problem (See App G.2) and showed how insights from our model can be applied to a real large-scale dataset for frequency estimation (see App G.1). These are standard datasets that prior work on approximate nearest neighbours and learning-augmented algorithms use to benchmark their methods on more realistic data.

**Generality of the Framework**

In our responses to reviewer mekY we discussed how our framework can be applied to new data structure problems. We believe there may be misunderstanding about the extent to which domain-specific expertise is needed to apply our framework to other problems. As we outline in our response, beyond transferring the high-level principles of our framework, we believe minimal architectural modifications are required to adapt to new problems (such as those we discuss in Section 3.2).

**Scaling/Computational Costs**

A common misunderstanding about our work was that we are proposing a system that should be used in production as a substitute for large scale database systems. While this is a worthwhile goal and something for which our work can serve as a starting point, our main focus is on using end-to-end learning to gain insight into data structure design, which we believe can be accomplished at small scale (e.g. our augmented CountMinSketch experiment in App G.1). We provided additional experiments showing how computationally cheaper alternatives (e.g. linear attention) can be substituted for the quadratic transformer we used to facilitate scaling to larger instances. Future work could explore this direction further. We will also include a more thorough discussion on limitations and avenues for scaling in our final revision.

Thank you for your consideration!

---

### Meta-Review · Area_Chair_Ke45 · 2024-12-19

**Metareview:**

Thanks for your submission to ICLR.

This paper received four reviews, three of which were on the negative side.  Many of the concerns in this paper revolved around the practicality and utility of the approach, the computational expense, concerns about what was actually discovered, and some issues with limited experimentation.  The positive reviewer felt fairly strongly about the promise of this approach, and was willing to champion the paper.  However, the other three reviewers, even after discussion, still were not enthusiastic enough to push for accepting the paper.

I appreciate the amount of work that the authors went into putting together the rebuttals and responses.  I tend to think that this approach has some promise, and I think with some additional work, this could make a very interesting paper indeed.  But I think that, in its current form, there simply isn't enough enthusiasm from the reviewers to warrant accepting this paper.

**Additional Comments On Reviewer Discussion:**

Several of the reviewers responded to the author rebuttal, with the most positive reviewer indicating that they disagreed with the negative reviews, but with one of the negative reviewers still maintaining a reject score.  The author responses did alleviate some concerns, but there are more fundamental issues here that still need to be addressed, including on the utility of the approach.

---

### Decision · Program_Chairs · 2025-01-22

Reject